# Coordinated regulation of vegetative phase change by brassinosteroids and the age pathway in *Arabidopsis*

Bingying Zhou [1,2,4], Qing Luo[1,2,4], Yanghui Shen[2], Liang Wei[2], Xia Song[2], Hangqian Liao[2], Lan Ni[3], Tao Shen[3], Xinglin Du[1], Junyou Han[1], Mingyi Jiang[3], Shengjun Feng [2] ✉ & Gang Wu [2] ✉

Vegetative phase change in plants is regulated by a gradual decline in the level of miR156 and a corresponding increase in the expression of its targets, *SQUAMOSA PROMOTER BINDING PROTEIN-LIKE* (*SPL*) genes. Gibberellin (GA), jasmonic acid (JA), and cytokinin (CK) regulate vegetative phase change by affecting genes in the miR156-SPL pathway. However, whether other phytohormones play a role in vegetative phase change remains unknown. Here, we show that a loss-of-function mutation in the brassinosteroid (BR) biosynthetic gene, *DWARF5* (*DWF5*), delays vegetative phase change, and the defective phenotype is primarily attributable to reduced levels of SPL9 and miR172, and a corresponding increase in TARGET OF EAT1 (TOE1). We further show that GLYCOGEN SYNTHASE KINASE3 (GSK3)-like kinase BRASSINOSTEROID INSENSITIVE2 (BIN2) directly interacts with and phosphorylates SPL9 and TOE1 to cause subsequent proteolytic degradation. Therefore, BRs function to stabilize SPL9 and TOE1 simultaneously to regulate vegetative phase change in plants.

After germination, higher plants undergo a juvenile and an adult phase of development before they acquire reproductively competence. The juvenile-to-adult phase transition is referred to as vegetative phase change. In *Arabidopsis*, vegetative phase change is marked by the production of abaxial trichomes on leaf blades, an increase in the leaf length-to-width (L/W) ratio, an increase in the degree of serration of the leaf margin, and a decrease in cell size[1–3].

Previous studies have shown that vegetative phase change is regulated by the evolutionarily conserved miR156-SPL pathway, also known as the age pathway, in plants, in which miR156 functions to promote juvenile development, whereas most of its targets, *SPL* genes, function to accelerate adult development. miR156 is highly accumulated in the juvenile phase, while its abundance decreases gradually as plants age. Correspondingly, the expression of *SPL* genes increases

during plant development[4–6]. Another miRNA, miR172, which exhibits a complementary expression pattern to miR156 during development, functions to promote vegetative phase change by repressing a class of *AP2*-like genes, including *APETALA2* (*AP2*), *TOE1*, *TOE2*, *TOE3*, *SCHLAFMUTZE* (*SMZ*), *SNARCHZAPFEN* (*SNZ*), and it is a direct transcriptional target of SPL9. miR172 and *AP2*-like genes act specifically to regulate leaf epidermal trait development by affecting the expression of *GLABRA1* (*GL1*) during vegetative phase change[5,7]. The Arabidopsis genome encodes sixteen *SPL* genes, among which ten are targeted by miR156. These ten *SPL* genes can be further divided into three groups based on their function in plant development. The first group, including *SPL2*, *SPL9*, *SPL10*, *SPL11*, *SPL13*, and *SPL15*, plays roles in both vegetative phase change and flowering;[8] *SPL3*, *SPL4*, and *SPL5* constitute the second group to regulate floral meristem identity;[9] *SPL6*,

[1]College of Plant Sciences, Jilin University, Jilin 130062, China. [2]The State Key Laboratory of Subtropical Silviculture, The Key Laboratory of Quality and Safety Control for Subtropical Fruit and Vege-table, Ministry of Agriculture and Rural Affairs, College of Horticultural Science, Zhejiang A&F University, Hangzhou 311300 Zhejiang, China. [3]College of Life Sciences, Nanjing Agricultural University, Nanjing, China. [4]These authors contributed equally: Bingying Zhou, Qing Luo. ✉e-mail: 20170039@zafu.edu.cn; wugang@zafu.edu.cn

belonging to the third group, may be critical for certain physiological processes[8].

miR156 represses *SPL* gene expression by transcript cleavage and translational repression[8,10]. In addition to being regulated by miR156, post translational modifications (PTMs) of the SPL proteins have also been shown to be crucial for their function. O-fucosyltransferase SPINDLY (SPY) interacts with SPL15 directly, and O-glycosylation of SPL15 by SPY inhibits SPL15 activity to regulate developmental transitions[11]. In rice, OsSPL14 or IDEAL PLANT ARCHITECTURE1 (IPA1), the homolog of Arabidopsis SPL9, interacts with IPA1 INTERACTING PROTIN1 (IPI1) and OsOTUB1 physically, leading to OsSPL14 ubiquitination and subsequent degradation to modulate rice architecture[12,13]. The phosphorylation status of Ser[163] in OsSPL14 also affects its function to balance growth and immunity in rice[14]. Previously, we identified some potential phosphorylation sites adjacent to the nuclear localization signal in the Arabidopsis SPL9 protein[15], but the function of SPL9 phosphorylation in plant development and how SPL9 is phosphorylated by upstream factors remain largely unknown.

Till now, three phytohormones, GA, JA, and CK have been shown to regulate vegetative phase change in *Arabidopsis*, maize, and rice[2,16–19]. The effect of GA on vegetative phase change is mediated by the DELLA proteins, a family of transcriptional repressors that are negatively regulated by GA[20]. In the absence of GA, the DELLA proteins interact with the SPL9 protein to interfere with its transcriptional activity[21]. JA promotes juvenile development possibly through activating the expression of miR156 in maize and rice via unknown mechanisms[17,18]. Cytokinin regulates vegetative phase change in *Arabidopsis* by affecting gene expression in the miR172/TOE1-TOE2 module downstream of the *SPL* genes[19].

BRs are a class of plant-specific steroid hormones that play vital roles in plant growth, development, and stress response[22,23]. BR is perceived by BRASSINOSTEROID INSENSITIVE1 (BRI) and its co-receptor kinase BRI1-ASSOCIATED RECEPTOR KINASE1 (BAK1) on the cell surface[24,25]. The BRI1 KINASE INHIBITOR1 (BKI1), a negative regulator of BRI1 signaling, then dissociates from the plasma membrane[26]. BRI and BAK1 sequentially phosphorylate and activate the BR signaling cascade[27]. Activated BRI1 phosphorylates two membrane-localized receptor-like cytoplasmic kinases, BRASSINOSTEROID-SIGNALING KINASE1 (BSK1) and CONSTITUTIVE DIFFERENTIAL GROWTH1/ CDG-LIKE (CDG1)[28,29], which in turn activates the nucleocytoplasmic phosphatase BRI1 SUPPRESSOR 1/BSU1-LIKE (BSU1/BSL)[30]. BSUI then dephosphorylates and inactivates BRASSINOSTEROID INSENSITIVE2 (BIN2), the GSK3-like kinase, leading to its subsequent degradation in a proteasome-dependent manner[31,32]. The inhibition of BRASSINAZOLE-RESISTANT1 (BZR1) and BRI1-EMS-SUPPRESSOR1 (BES1) or BZR2 by BIN2 is then released, and they are translocated to the nucleus to regulate downstream target genes[33]. Although BRs have been shown to involve a plethora of different biological processes, whether they play a role in vegetative phase change still remains unknown.

In an attempt to identify more players in the miR156-SPL pathway to regulate vegetative phase change, we performed a forward genetic screen in an M₂ population from an ethyl methanesulfonate (EMS)-mutagenized Arabidopsis Columbia-0 (Col-0) seeds. In this work, we identified a mutant exhibiting a delayed vegetative phase change phenotype, which we named *delayed juvenile-to-adult phase transition mutant2* (*del2*). Map-based cloning indicated that *del2* has a mutation in the *DWF5* gene encoding the Δ7-sterol reductase, which functions to convert 5-dehydroepisterol to 24-methylenecholesterol (24-MC) in the BR biosynthetic pathway[34]. Our results indicate that the defective phenotype of *dwf5* is mainly attributable to its downregulation of the SPL9 protein and miR172, and a corresponding upregulation of the TOE1 protein. Molecular and biochemical experiments demonstrated that BIN2 physically interacts with SPL9 and TOE1 in vitro and in vivo, and phosphorylates SPL9 and TOE1 to cause their subsequent proteolytic degradation and functional sequestration. Therefore, our

study identified a PTM pattern in the SPL9 and TOE1 proteins by BIN2 necessary for SPL9 and TOE1 stabilization and normal function, and reveals an unidentified role of the BR signaling pathway in vegetative phase change by interacting with the age pathway.

## Results

### A mutant defective in BR biosynthesis exhibits a delayed vegetative phase change phenotype

The miR156-SPL age pathway is the master regulatory pathway controlling vegetative phase change in plants[5]. However, how different pathways integrate into the age pathway to regulate vegetative phase change remains largely unexplored in plants. In an attempt to identify more factors involved in this pathway, we screened a selfed M₂ generation from an EMS-mutagenized wild-type (WT) Col-0 Arabidopsis population. We identified a mutant exhibiting a delayed vegetative phase change phenotype, which we initially named *del2*. In short days, WT plants produced abaxial trichomes on leaf 8.3, whereas *del2* produced significantly later abaxial trichomes on leaf 10.8 (Fig. 1a). Compared with WT, *del2* had a significantly slower leaf initiation rate (Fig. 1b), and its leaves were also much rounder and smaller (Fig. 1a, c), typical characteristics of juvenile traits. In addition, *del2* also exhibited pleiotropic defects, including an overall smaller and dwarfed stature with dark green leaves, and developmental retardation (Fig. 1a–c).

We crossed *del2* to Landsberg *erecta* (Ler) to generate a segregating F₂ population to map the *del2* mutation. We narrowed down the *del2* mutation to a region between markers F14I3 and F11F12 on chromosome I (Fig. 1d). We started to focus on AT1G50430 (*DWF5*), a gene required for BR biosynthesis, in this region because mutants in the *DWF5* gene are usually smaller in overall stature with dark green leaf color[34]. Therefore, we sequenced the *DWF5* gene in *del2*. The sequencing result indicated that there was a G-to-A substitution at the 1848th base in the *DWF5* coding region, which resulted in a replacement of the tryptophon (TGG) by a stop codon (TGA) to cause an early translational termination of the DWF5 protein (Fig. 1e).

To verify if this mutation is responsible for the *del2* defective phenotype, we amplified the genomic sequence of *DWF5* containing the promoter, coding, and 3' sequence, and cloned it into an expression vector to generate the *pDWF5::DWF5* construct, and transformed the *del2* plants. Phenotypic characterization of different primary transformants in short days indicated that the transgenic plants in the *del2* background had indistinguishable phenotypes from WT (Fig. 1a). This result suggests that the defective *del2* phenotype is truly attributable to the mutation in the *DWF5* gene. Therefore, we renamed this mutation to *dwf5*.

Since *dwf5* is defective in BR biosynthesis, we determined the level of endogenous BRs in *dwf5*. As expected, the content of brassinolide (BL) in *dwf5* was significantly lower than that in WT (Fig. 1f). Then we asked if exogenous addition of BR can restore the *dwf5* phenotype. We treated WT and *dwf5* seedlings with different concentrations of BL, and characterized their vegetative phase change phenotype in short days. 10 nM BL treatment had almost no effect on WT phenotypes, whereas BL treatment of *dwf5* plants slightly but significantly accelerated the abaxial trichome production (*dwf5*-MOCK 10.2 ± 0.6 versus *dwf5*−10nM BL 9.0 ± 0.7). This result suggests that *dwf5* defective phenotype is partially due to the reduced level of BL in plants (Fig. 1g). The inability of BL treatment to fully restore the *dwf5* defective phenotype is probably due to the low absorption of exogenously applied BL by plants or other unknown mechanisms. In the case of WT, it is possible that WT already accumulates enough BRs required for normal development, a further increase in BRs will not alter its normal development.

### *dwf5* affects the stability and function of SPL9 instead of transcripts of genes in the miR156-SPL pathway

Next, we asked if the delayed vegetative phase change phenotype of *dwf5* is attributable to changes in the expression of genes in the

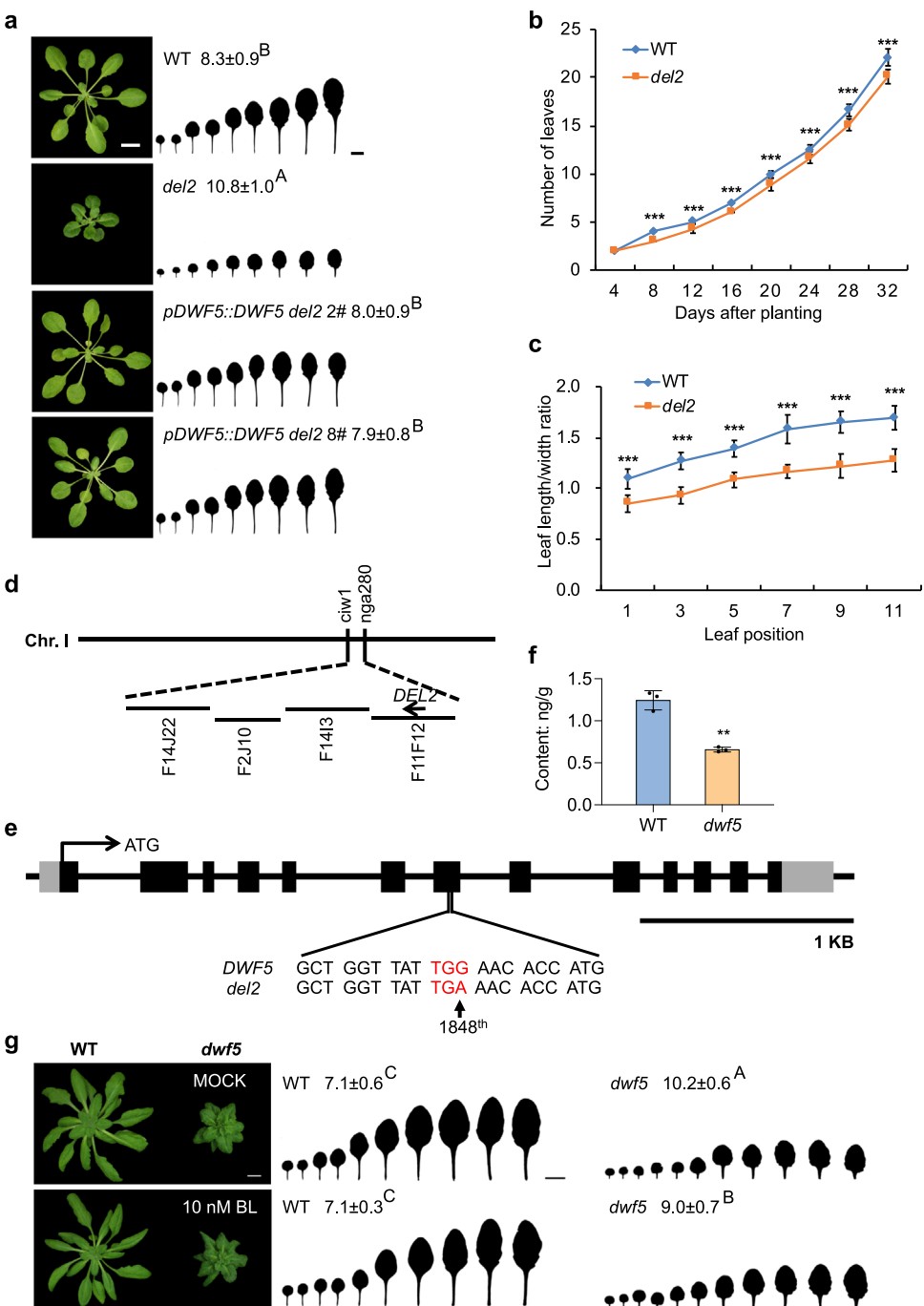

**Fig. 1 | Loss-of-function mutation in BR biosynthesis delays vegetative phase change in *Arabidopsis*. a** Phenotypic characterization of the *del2* mutant. 28-day-old wild type (WT, Col-0), *del2*, *pDWF5::DWF5 del2 2#, 8#* transgenic lines were grown in short days. The first leaf with abaxial trichomes was scored. Numbers indicate the first leaf with abaxial trichomes (*n* = 25 plants, ±SD). Different letters indicate significant difference between genotypes using one-way ANOVA at *P* < 0.001. Scale bar = 1 cm. **b** Leaf initiation rate of WT and *del2* in short days. Leaf numbers were scored at 4, 8, 12, 16, 20, 24, 28, 32 DAP. DAP, days after planting. Asterisks denote significant difference from WT using two-tailed Student's *t*-test (*P* < 0.001, *n* = 30 plants, ±SD). **c** The length-to-width (L/W) ratios of leaf 1-11 from 45-day-old WT and *del2* in short days. Asterisks denote significant difference from WT using two-tailed Student's *t*-test (*P* < 0.001, *n* = 10 plants, ±SD). **d** The chromosomal location of the *del2* mutation. The *del2* mutation was narrowed down to a region between markers ciw1 and nga280 on chromosome I. **e** The mutation site in *dwf5*. Sequencing of *DWF5* identified a G-to-A substitution at the 1848th base (shown by the arrow) in the coding region, which caused an early translational termination of the *DWF5* gene. Scale bar = 1 KB. **f** Quantitation of BL levels in 15-day-old WT and *dwf5*. Asterisks denote significant difference from WT using two-tailed Student's *t*-test (*P* < 0.01, *n* = 3 biologically independent samples, ±SD). **g** Phenotypic characterization of MOCK- and BL-treated 30-day-old WT and *dwf5*. WT and *dwf5* were initially grown on 1/2 MS plates containing 0 or 10 nM BL in short days for 10 days, then transferred to 1/2 hogland liquid medium with 0 or 10 nM BL in short days. Numbers indicate the first leaf with abaxial trichomes (*n* = 15 plants, ±SD). Different letters indicate significant difference between genotypes using one-way ANOVA at *P* < 0.001. Scale bar = 1 cm. All experiments were repeated 3 times biologically.

miR156-SPL pathway. We measured the expression of some key genes in the miR156-SPL pathway using RNA from 12-day-old seedlings of WT and *dwf5* mutant in short days. Quantitative reverse transcription-PCR (qRT-PCR) revealed that levels of mature miR156, *SPL3*, *SPL9*, *SPL13*, *TOE1*, and *TOE2* were similar between WT and *dwf5* (Fig. 2a, b). However, the abundance of *MIR172B* and mature miR172 was both reduced remarkably in *dwf5* (Fig. 2b). It is not surprising to see that the expression of the downstream *TOE1/TOE2* in *dwf5* remained indistinguishable from that in WT because miR172 represses *AP2-like* genes mainly by translational repression[35]. These results imply that the delayed vegetative phase change phenotype of *dwf5* might result from the reduced levels of miR172.

Since SPL9 functions as a direct activator of *MIR172B*[5], we asked if the level or the function of SPL9 were affected in *dwf5*. We first determined the level of the SPL9 protein in *dwf5*. Due to the lack of available SPL9 antibodies, we crossed the *pSPL9::3×FLAG-rSPL9* transgenic line in which a 3×FLAG epitope tag was fused to the miR156 insensitive form of SPL9 under the regulation of its native promoter to *dwf5*[5]. Using the FLAG antibody, we performed Western blotting analysis to examine the level of the SPL9 protein in the WT and *dwf5* background. Western blotting indicated a >4-fold reduction in the SPL9 protein level in *dwf5* compared with that in WT (Fig. 2c), suggesting that SPL9 was destabilized in *dwf5*. We further explored the effect of *dwf5* on the function of SPL9 by using an inducible expression system based on the posttranscriptional activation of the rat glucocorticoid receptor (GR)[36]. GR was fused to the 5' end of miR156 insensitive SPL9 (*rSPL9*), and this fusion gene was expressed in transgenic plants under the regulation of its native promoter. *pSPL9::GR-rSPL9* was crossed to *dwf5*, and homozygous *pSPL9::GR-rSPL9 dwf5* plants were recovered in the F$_2$ population. Transgenic *pSPL9::GR-rSPL9* seeds in WT and *dwf5* background were plated on 1/2 MS medium, and treated with the synthetic ligand dexamethasone (DEX) or MOCK for 3 h. RNA was extracted and the abundance of *MIR172B* was assessed by qRT-PCR. As expected, *MIR172B* expression was significantly induced by about 2.6-fold upon DEX induction in WT background compared with an about 1.5-fold induction in the *dwf5* background (Fig. 2d). Therefore, the incapacity to fully activate the expression of *MIR172B* demonstrates that the function of SPL9 is partially impaired in *dwf5*, and *DWF5* integrates into the miR156-SPL pathway by stabilizing the SPL9 protein and maintaining its function.

To further understand if the reduced level of the SPL9 protein is attributable to the reduced stability of SPL9 in *dwf5*, we treated *pSPL9::3×FLAG-rSPL9* plants with MG132 and/or cycloheximide (CHX). As expected, the SPL9 protein level was lower in *dwf5* than that in WT in the DMSO-treated samples, CHX treatment alone led to a further reduction in SPL9 in *dwf5* than in WT. However, MG132 treatment along or in combination with CHX inhibited SPL9 degradation (Fig. 2e). These results suggest that SPL9 is subjected to proteasome-dependent degradation, and it is much less stable in *dwf5* than in WT. Therefore, BRs function to stabilize SPL9 in vegetative phase change. This conclusion is also further supported by the result that the addition of BL to *pSPL9::3×FLAG-rSPL9 dwf5* plants increased the level of SPL9 greatly at different time points (Fig. 2f).

### Downregulation of miR172 is partially responsible for the *dwf5* vegetative phase change phenotype

We explored the possibility of if the reduced level of miR172 is responsible for the delayed vegetative phase change phenotype in *dwf5*. miR172 represses the expression of a class of *AP2-like* genes, including *TOE1*, *TOE2*, *TOE3*, *AP2*, *SMZ*, and *SNZ*. Plants doubly mutant for *TOE1* and *TOE2* resemble the phenotype of miR172 overexpression line[5]. Even though levels of *TOE1/TOE2* transcripts remained similar between WT and *dwf5* (Fig. 2b), it is possible that the TOE1/TO2 proteins levels are elevated in *dwf5* due to the reduced level of miR172 that functions to repress *AP2*-like gene expression through translational

repression. Therefore, we crossed the miR172 over-expression line *Ubi10::172B* and *toe1 toe2* double mutant to *dwf5*, respectively. We then characterized the phenotypes of homozygous *Ubi10::172B dwf5*, *toe1 toe2 dwf5*, and their corresponding parental lines in short days. *dwf5* produced late abaxial trichomes on leaf 11.3, *Ubi10::172B* and *toe1 toe2* produced early abaxial trichomes on leaf 3.8 and 3.6, respectively; whereas *Ubi10::172B dwf5* and *toe1 toe2 dwf5* produced abaxial trichomes on leaf 4.5 and 3.8 (Fig. 3a), significantly earlier than *dwf5*. *toe1 toe2* was almost completely epistatic to *dwf5* with respect to abaxial trichome production. Restoration of miR172 expression in the *dwf5* background could also significantly rescue the late vegetative phase change phenotype of *dwf5*. These results, together with the downregulation of miR172 in *dwf5*, suggest that miR172 functions downstream of *DWF5* with respect to leaf epidermal trait development during vegetative phase change, and the reduction of miR172 in *dwf5* is partially responsible for the late abaxial trichome phenotype of *dwf5*. A gradual change in leaf shape constitutes an important morphological marker associated with vegetative phase change. In contrary to distinct abaxial trichome phenotypes as manifested by *dwf5*, *Ubi10::172B dwf5*, and *toe1 toe2 dwf5*, leaf shape of *Ubi10::172B dwf5* and *toe1 toe2 dwf5* resembled that of *dwf5* to a greater extent. This result suggests that *dwf5* is epistatic to miR172 and *TOE1/TOE2* with respect to leaf shape development during vegetative phase change, and BRs contribute to leaf shape development by affecting genes other than miR172 and *TOE1/TOE2*. This is also in accordance with our previous result that miR172 and *TOE1/TOE2* only contribute to leaf epidermal trait development in vegetative phase change[5].

To test the significance of the downregulation of SPL9 in *dwf5*, we crossed the *pSPL9::rSPL9* line overexpressing a miR156-resistant version of *SPL9* under the control of its native promoter to *dwf5*. We then characterized the vegetative phase change phenotype of the homozygous *pSPL9::rSPL9 dwf5* line in short days. On average, *pSPL9::rSPL9 dwf5* produced slightly but not significantly later abaxial trichomes than *pSPL9::rSPL9* (1.4 ± 0.7 versus 1.0 ± 0.0), but remarkably earlier than *dwf5* did (Fig. 3a). A detailed phenotypic characterization of *pSPL9::rSPL9 dwf5* plants revealed an interesting phenotype: 100% of *pSPL9::rSPL9* plants produced abaxial trichomes on leaf 1, whereas 75.7%, 15.7%, and 8.6% of *pSPL9::rSPL9 dwf5* plants produced abaxial trichomes on leaf 1, 2, and 3, respectively (Fig. 3b). Moreover, *pSPL9::rSPL9 dwf5* plants also had a significantly reduced number of abaxial trichomes on the first leaf than *pSPL9::rSPL9* plants (Fig. 3c). Last but not least, the first leaves of *pSPL9::rSPL9 dwf5* plants had a significantly smaller L/W ratio than those of *pSPL9::rSPL9* plants, which is similar to that of WT (Fig. 3d). These genetic data demonstrate that *DWF5* is required for *SPL9* to promote vegetative phase change, and the function of SPL9 is sequestered in *dwf5*, which is consistent with the molecular data as shown in Fig. 2d. We also generated double mutant between *dwf5* and *spl9-4*. *spl9-4 dwf5* had an enhanced vegetative phase change phenotype than *spl9-4* and *dwf5* with respect to abaxial trichome production (Fig. 3a). Together with the result that SPL9 is reduced in *dwf5*, these results suggest that *DWF5* functions to regulate vegetative phase change through SPL9 dependent and independent pathways.

### BIN2 physically interacts with SPL9

BIN2, a GSK3-like kinase, acts as one of the central hubs in the BR signaling pathway to negatively regulate BR signaling by interacting with and phosphorylating its substrates[37–39]. To investigate if the reduced level of SPL9 in *dwf5* is mediated by BIN2, we first tested if SPL9 interacts with BIN2 using a yeast two-hybrid system. We fused the full-length of *SPL9* coding sequence (CDS) to the GAL4 activation domain to generate the prey vector *AD-SPL9*, and fused the *BIN2* CDS to the GAL4 DNA binding domain to generate the bait vector *BD-BIN2*. These two vectors were then co-transformed into the yeast competent cells, and plated on DDO (SD/-Leu/-Trp) and QDO (SD/-Leu/-Trp/-Ade/-His) media. As shown in Fig. 4a, BIN2 interacted with SPL9 in the yeast

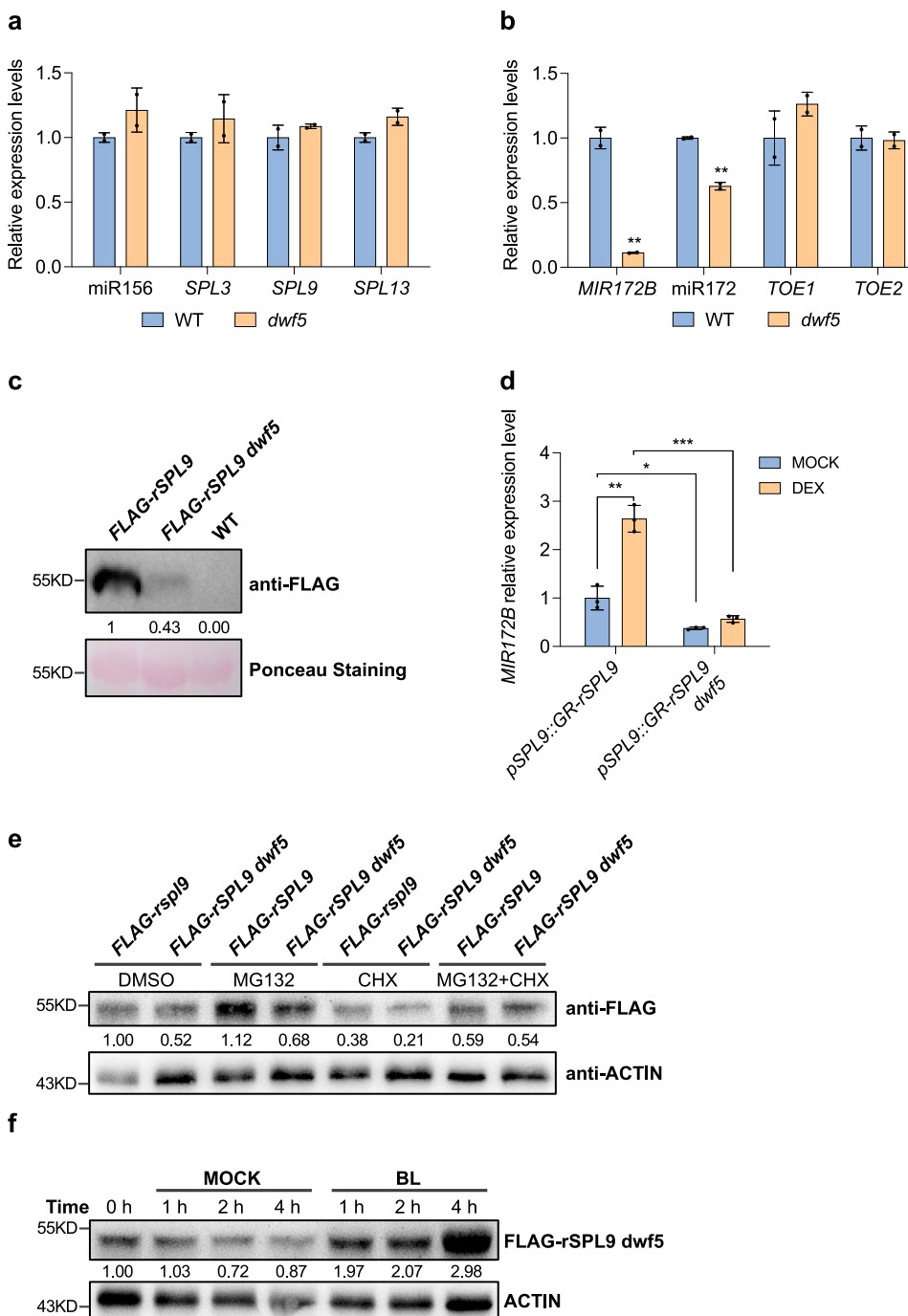

**Fig. 2 | *dwf5* destabilizes the SPL9 protein and sequesters its function instead of affecting miR156 and *SPL* gene expression. a**, **b** The expression levels of miR156, *SPL3*, *SPL9*, *SPL13* (**a**), *MIR172B*, miR172, *TOE1* and *TOE2* (**b**) in 12-day-old WT and *dwf5* in short days. Data are means ± SD from a representative experiment with two technical replicates for each sample. Asterisk denotes significant difference from WT using Student's *t*-test at *P* < 0.01. **c** *dwf5* reduces the SPL9 protein level. The SPL9 protein in *pSPL9::3×FLAG-rSPL9* and *pSPL9::3×FLAG-rSPL9 dwf5* transgenic lines was detected by Western blotting using an anti-FLAG antibody. The Rubisco protein was stained with ponceau as the protein loading control, WT was served as the negative control. Numbers between two blots denote the relative normalized value for each sample. **d** *dwf5* sequesters SPL9 function to activate the expression of *MIR172B*. 15-day-old *pSPL9::GR-rSPL9* and *pSPL9::GR-rSPL9 dwf5* plants grown on 1/2 MS medium in short days were treated with MOCK or dexamethasone (DEX) for 3 h, and total RNA was isolated for qRT-PCR analysis. Data are means ± SD from a representative experiment with three technical replicates for each sample. Asterisk

denotes significant difference using one-way ANOVA; **p* < 0.05; ***P* < 0.01; ****P* < 0.001. **e** Destabilization of the SPL9 protein in *dwf5* is proteasome-dependent. 15-day-old *pSPL9::3×FLAG-rSPL9* and *pSPL9::3×FLAG-rSPL9 dwf5* plants were treated with 50 μM MG132 and/or 100 mM CHX for 1 h in short days. The total protein was extracted and detected by Western blotting using an anti-FLAG and anti-ACTIN antibody, respectively. **f** BL treatment of *dwf5* increases SPL9 accumulation. 15-day-old *pSPL9::3×FLAG-rSPL9 dwf5* plants in short days were treated with MOCK or 1 μM BL for 1 h, and total protein was extracted and detected by Western blotting using an anti-FLAG and anti-ACTIN antibody, respectively. Numbers between two blots denote the relative normalized value for each sample. The intensity of each sample was first normalized to the Rubisco band or the corresponding ACTIN band (**c**, **e**, **f**), then the resultant value was normalized again to the value of *FLAG-rSPL9* (**c**), *FLAG-rSPL9* DMSO (**e**), and *FLAG-rSPL9 dwf5* 0 h (**f**). Band intensity was determined using Image J. All experiments were repeated 3 times biologically.

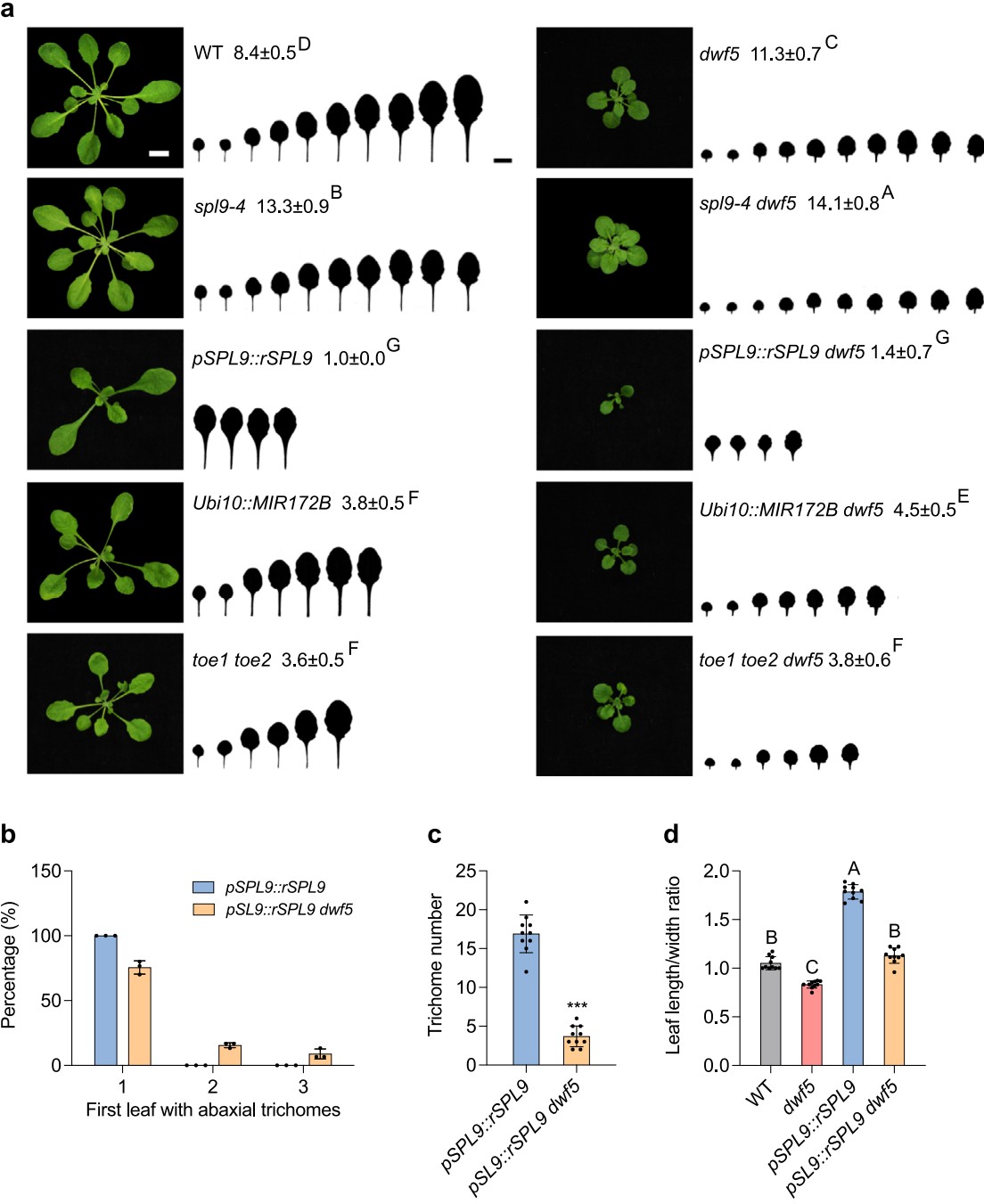

**Fig. 3 | Genetic interaction of *SPL9* and miR172 with *dwf5*. a** Phenotype of 25-day-old WT, *dwf5*, *spl9-4*, *spl9-4 dwf5*, *pSPL9::rSPL9*, *pSPL9::rSPL9 dwf5*, *Ubi10::MIR172B*, *Ubi10::172B dwf5*, *toe1 toe2*, and *toe1 toe2 dwf5* plants grown in short days. Numbers indicate the first leaf with abaxial trichomes (*n* = 25 plants, ±SD). Different letters indicate significant difference between genotypes using one-way ANOVA at *P* < 0.01. Scale bar = 1 cm. **b** The percentage of leaf position with abaxial trichomes in *pSPL9::rSPL9* and *pSPL9::rSPL9 dwf5* (*n* = 90 plants, ±SD). **c** Abaxial trichome number on the first leaf of *pSPL9::rSPL9* and *pSPL9::rSPL9 dwf5*. Asterisks denote significant difference from *pSPL9::rSPL9* using two-tailed Student's *t*-test at *P* < 0.001 (*n* = 10 plants, ±SD). **d** Leaf length/width ratio of the first leaf from 45-day-old WT, *dwf5*, *pSPL9::rSPL9*, and *pSPL9::rSPL9 dwf5*. Different letters indicate significant difference between genotypes using one-way ANOVA at *P* < 0.01 (*n* = 10 plants, ±SD). All experiments were repeated 3 times biologically.

two-hybrid system. We then performed a bimolecular fluorescence complementation (BiFC) assay to examine if BIN2 interacts with SPL9 in vivo. BIN2 was fused to the C-terminal of yellow fluorescent protein (YFP) (BIN2-cYFP), and SPL9 was fused to the N-terminal of YFP (SPL9-nYFP). We co-transformed these two vectors into WT protoplast isolated using a Tape-*Arabidopsis* Sandwich method[40]. YFP fluorescence was observed in the transformed cell nuclei overlapped with the nuclei stained by the Hoechst 33342 solution, whereas no fluorescence was detected in the negative control samples (Fig. 4b). This result

demonstrated that BIN2 and SPL9 interact with each other in vivo in Arabidopsis cells. To confirm if BIN2 interacts with SPL9 in vitro, we performed a protein pull-down assay. We first expressed the Glutathione S-Transferase (GST), GST-BIN2, and 6×His-SPL9 proteins in *E. coli*, then we used glutathione-agarose beads to bind the GST and GST-BIN2 proteins, and incubated them with His-SPL9 for 4 h in vitro. Last, we collected and boiled the beads to perform Western blotting using an anti-His and an anti-GST antibody, respectively. Western blotting indicated that BIN2 also interacts with SPL9 in vitro (Fig. 4c). As a

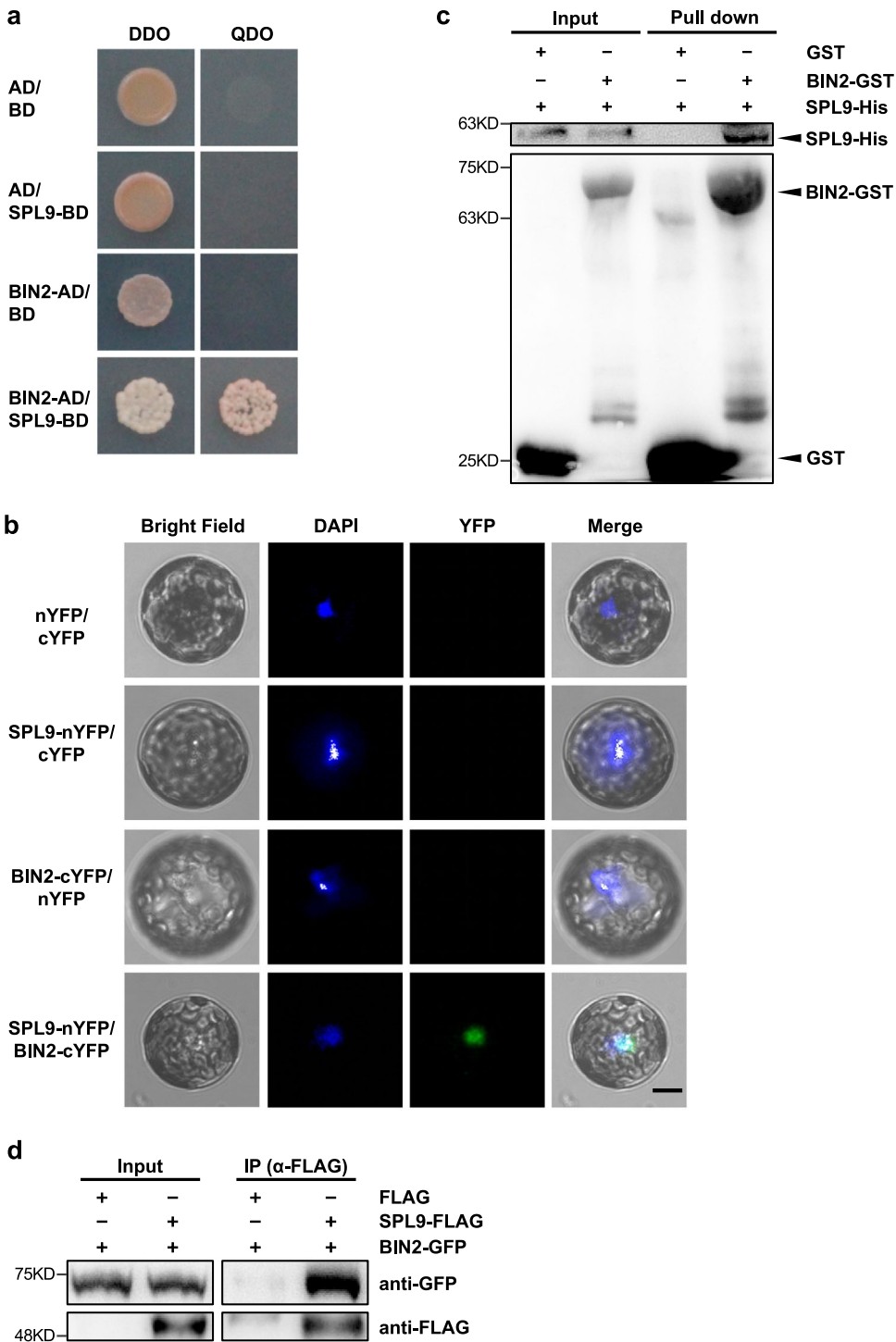

**Fig. 4 | BIN2 physically interacts with SPL9. a** BIN2 interacts with SPL9 in yeast. BIN2 was fused to pGAD (AD), and SPL9 was fused to pGBK (BD). AD and BD empty vectors were used as the negative controls. DDO (SD/-Leu/-Trp), QDO (SD/-Leu/-Trp/-Ade/-His). **b** BIN2 interacts with SPL9 in vivo in the nucleus as shown by bimolecular fluorescence complementation (BiFC) assay. BIN2 was fused to cYFP, and SPL9 was fused to nYFP. The recombinant plasmids were transformed into the WT Arabidopsis protoplast. The nucleus was stained with hoechest 33342. Scale bar = 10 μm. **c** BIN2 interacts with SPL9 in vitro in a pull-down assay. SPL9-His was incubated with GST or GST-BIN2 protein, and the proteins immunoprecipitated with glutathione-agarose beads were detected with an anti-His and an anti-GST antibody, respectively. **d** BIN2 interacts with SPL9 in vivo in an CoIP assay. Total protein was extracted from *Nicotiana benthamiana* leaves infiltrated with *Agrobacterium* containing *35 S::3×FLAG/35 S::3×FLAG-rSPL9*, and *35 S::BIN2-GFP/ 35 S::3×FLAG-rSPL9* combinations, and incubated with anti-FLAG beads. Coimmunoprecipitated proteins were detected with an anti-FLAG and an anti-GFP antibody, respectively. All experiments were repeated 3 times biologically.

further step to confirm the interaction between BIN2 and SPL9 *in planta*, we performed a coimmunoprecipitation (CoIP) assay. We co-expressed SPL9-FLAG and BIN2-GFP fusion proteins in tobacco (*Nicotiana benthamiana*) leaves, and incubated the total protein with protein affinity gel conjugated with the anti-FLAG antibody. Then we performed Western blotting to detect the eluate with an anti-FLAG and an anti-GFP antibody, respectively (Fig. 4d). The result showed that SPL9 and BIN2 were coimmunoprecipitated. Taken together, our results demonstrate that BIN2 physically interacts with SPL9 both in vitro and in vivo.

We then asked if the reduced level of SPL9 in *dwf5* is due to the action of the elevated level of BIN2 since *dwf5* alleles are defective in the $\Delta^7$ reduction step in the BR biosynthetic pathway[34]. We first examined if the level of BIN2 is elevated in *dwf5*. To do this, we fused the BIN2 CDS sequence to eGFP, and put it under the control of the BIN2 native promoter to generate the *pBIN2::eGFP-BIN2* construct. We then transformed WT, and generated a stable transgenic *pBIN2::eGFP-BIN2* line with a single T-DNA insertion in the progeny. We also crossed this line to *dwf5* to generate a homozygous *pBIN2::eGFP-BIN2 dwf5* line. Western blotting using an anti-GFP antibody indicated that the BIN2 protein level was elevated in the *dwf5* background in contrast to the WT background (Supplementary Fig. 1a). To further understand if the elevated level of BIN2 contributes to the *dwf5* vegetative phase change phenotype, we germinated WT and *dwf5* seeds on 1/2 MS medium supplemented with 10 mM LiCl, a GSK3-like inhibitor[41,42], and characterized their corresponding phenotypes in short days. Both mock and LiCl-treated WT produced abaxial trichomes on about leaf 8.0, whereas LiCl treatment significantly accelerated the production of abaxial trichomes in *dwf5* (LiCl 9.4 ± 0.7 versus Mock 11.7 ± 0.7) (Supplementary Fig. 1b). These results suggest that the delayed vegetative phase change phenotype in *dwf5* is partially attributable to the elevated level of the BIN2 protein.

## BIN2-mediated phosphorylation of SPL9 is required for its function in vegetative phase change

Since BIN2 physically interacts with SPL9, it is possible that BIN2 phosphorylates SPL9 to cause its subsequent degradation or functional sequestration. Protein sequence analysis of SPL9 uncovered a GSK3 kinase classical phosphorylation motif of (T/S)-X-X-X-(T/S) (T/S denotes Thr/Ser; X denotes any other amino acids) in the conserved SBP domain (Fig. 5a). To investigate if SPL9 is phosphorylated by BIN2 in vitro, we purified the bacterium-expressed GST-BIN2 and 6×His-SPL9 fusion proteins, and conducted an in vitro kinase assay in an SDS-PAGE gel containing the phos-tag reagent. Western blotting using an anti-His antibody showed a lagged 6×His-SPL9 band in the phos-tag gel (Fig. 5b), suggesting that SPL9 was phosphorylated by BIN2 in vitro. To robustly establish that SPL9 is a substrate of BIN2, we performed an in vitro kinase assay. We purified the bacterium-expressed GST-BIN2, 6×His-SPL9 (T) protein in which the 102nd and 106th amino acids are the wild-type threonine, and 6×His-SPL9 (A) in which both the 102nd and 106th threonines were replaced with an alanine. When GST-BIN2 was incubated with 6×His-SPL9 (T) and radiolabeled ATP, autoradiographs showed two bands with the top band corresponding to the auto-phosphorylated GST-BIN2 and the lower band corresponding to the phosphorylated His-SPL9 (Fig. 5c). However, when His-SPL9 (A) was incubated with GST-BIN2, the lower band corresponding to His-SPL9 in the autoradiograph almost disappeared (Fig. 5c). This result further demonstrates that SPL9 is a substrate of BIN2, and the motif of T-P-K-V-T in the SPL9 protein is critical for its phosphorylation by BIN2.

To investigate the function of SPL9 phosphorylation by BIN2 in vegetative phase change, we generated constructs expressing a miR156-insensitive form of *pSPL9::3×FLAG-rSPL9* (*rSPL9-TPKVT*) with a wild-type BIN2 classical phosphorylation form, a mutated BIN2 non-phosphorylation form of *pSPL9::3×FLAG-rSPL9* (*rSPL9-APKVA*), and a constitutive BIN2 phosphorylation form of *pSPL9::3×FLAG-rSPL9* (*rSPL9-DPKVD*) under the regulation of the SPL9 native promoter, and transformed these constructs into Col-0 WT. We first characterized the vegetative phase change phenotype of >70 different primary transgenic lines from each construct in short days. About 96% of *rSPL9-APKVA*, 65% of *rSPL9-TPKVT*, and 33% of *rSPL9-DPKVD* plants produced abaxial trichomes on leaf 1, respectively (Fig. 5d). Furthermore, about 38% of *rSPL9-DPKVD* plants produced abaxial trichomes on leaves later than 4, which was significantly higher than 4% in *rSPL9-APKVA* and 14% in *rSPL9-TPKVT*, suggesting that the constitutive phosphorylation form of SPL9 significantly impaired its

function, whereas non-phosphorylation form of SPL9 significantly enhanced its function to promote vegetative phase change. To understand how SPL9 phosphorylation affects vegetative phase change phenotype and the protein levels, we first generated homozygous lines with a single T-DNA insertion in progenies from these transgenic lines, and determined the transcript level of SPL9 in different homozygous lines. We then characterized the phenotype of these lines with comparable SPL9 transcript levels (Fig. 5e, f). Among them, T-A 2# and T-A 6# accumulated relatively more SPL9 protein than T-T 1# and T-T 10#, whereas T-D 4#, T-D 8# and T-D 50# had the lowest level of the SPL9 protein (Fig. 5g), which is also consistent with the phenotypic characterization result that the *rSPL9-DPKVD* transgenic plants produced much later abaxial trichomes on leaves 3–5 with much rounder first leaves (Fig. 5e, h). Within *rSPL9-APKVA* (*T-A*) and *rSPL9-TPKVT* (*T-T*) transgenic plants that produced abaxial trichomes on leaf 1, *rSPL9-APKVA* (*T-A*) produced the first leaves with a slightly bigger L/W ratio than *rSPL9-TPKVT* (*T-T*) did (Fig. 5e, h). These results imply that BIN2-mediated phosphorylation of the TPKVT motif in the conserved SBP domain destabilizes the SPL9 protein and sequesters its function, and this mode of regulation is critical for SPL9 function to promote vegetative phase change.

To understand how SPL9 is phosphorylated by BIN2 in vivo, we performed a liquid chromatography-mass spectrometry (LC-MS/MS) analysis. We first transformed the wild-type Wassilewskija (Ws) and the *bin2-3bil1bil2* mutant protoplast with the *pSPL9::3×FLAG-rSPL9* construct, then we immunoprecipitated the SPL9 protein and analyzed the status of SPL9 phosphorylation with LC-MS/MS. LC-MS/MS indicated that the 102nd threonine was phosphorylated in Ws-0, but not in the *bin2-3bil1bil2* mutant (Supplementary Fig. 2a). Interestingly, the 102nd threonine is exactly located in the GSK3 kinase classical phosphorylation motif of T-P-K-V-T in the conserved SBP domain (Fig. 5a, Supplementary Fig. 2a)[15,39]. Since BIN2 functions to destabilize SPL9, we expect that loss-of-function mutations in the BIN2 family members would result in an elevated level of the SPL9 protein. Therefore, we determined the level of SPL9 in *bin2-3 bil1 bil2* and WT using the protoplast transformation method. As expected, the level of SPL9 was significantly elevated by about 2.5-fold in *bin2-3 bil1 bil2* in contrast to that in WT (Supplementary Fig. 2b), this result is also consistent with the LC-MS/MS analysis result that the T-P-K-V-T motif in SPL9 was phosphorylated in WT, but not in *bin2-3 bil1 bil2* (Supplementary Fig. 2a).

## BIN2 physically interacts with TOE1

In short days, the *bin2-3 bil1 bil2* mutant produced abaxial trichomes on leaf 10.0, which was significantly later than WT with abaxial trichomes on leaf 6.6 (Supplementary Fig. 3a). This result seems to be contradictory to the elevated level SPL9 in *bin2-3 bil1 bil2* since SPL9 functions to promote vegetative phase change. This puzzle prompted us to ask if BIN2 also functions to regulate genes other than *SPL9* to regulate vegetative phase change. Because loss-of-function mutations in *TOE1* and *TOE2* could almost fully restore the late abaxial trichome phenotype of *dwf5* (Fig. 3a); therefore, we started to investigate if BIN2 could also interact with TOE1.

We first analyzed the TOE1 protein sequence. Interestingly, there is a GSK3 classical phosphorylation motif of T-K-L-V-T in TOE1 (Fig. 6a). We then tested if TOE1 physically interacts with BIN2 using the yeast two-hybrid system. We generated the prey vector AD-TOE1 and the bait vector BD-BIN2, and co-transformed into the yeast competent cells, and plated on DDO and QDO media. As shown in Fig. 6b, yeast cells only grew on the QDO media when AD-TOE1 and BD-BIN2 were co-transformed, suggesting that BIN2 interacted with TOE1 in the yeast two-hybrid system. We then performed a BiFC assay to examine if BIN2 also interacts with TOE1 in vivo. BIN2 was fused to the C-terminal of YFP (BIN2-cYFP), and TOE1 was fused to the N-terminal of YFP (TOE1-nYFP). We co-transformed different combinations of vectors into the

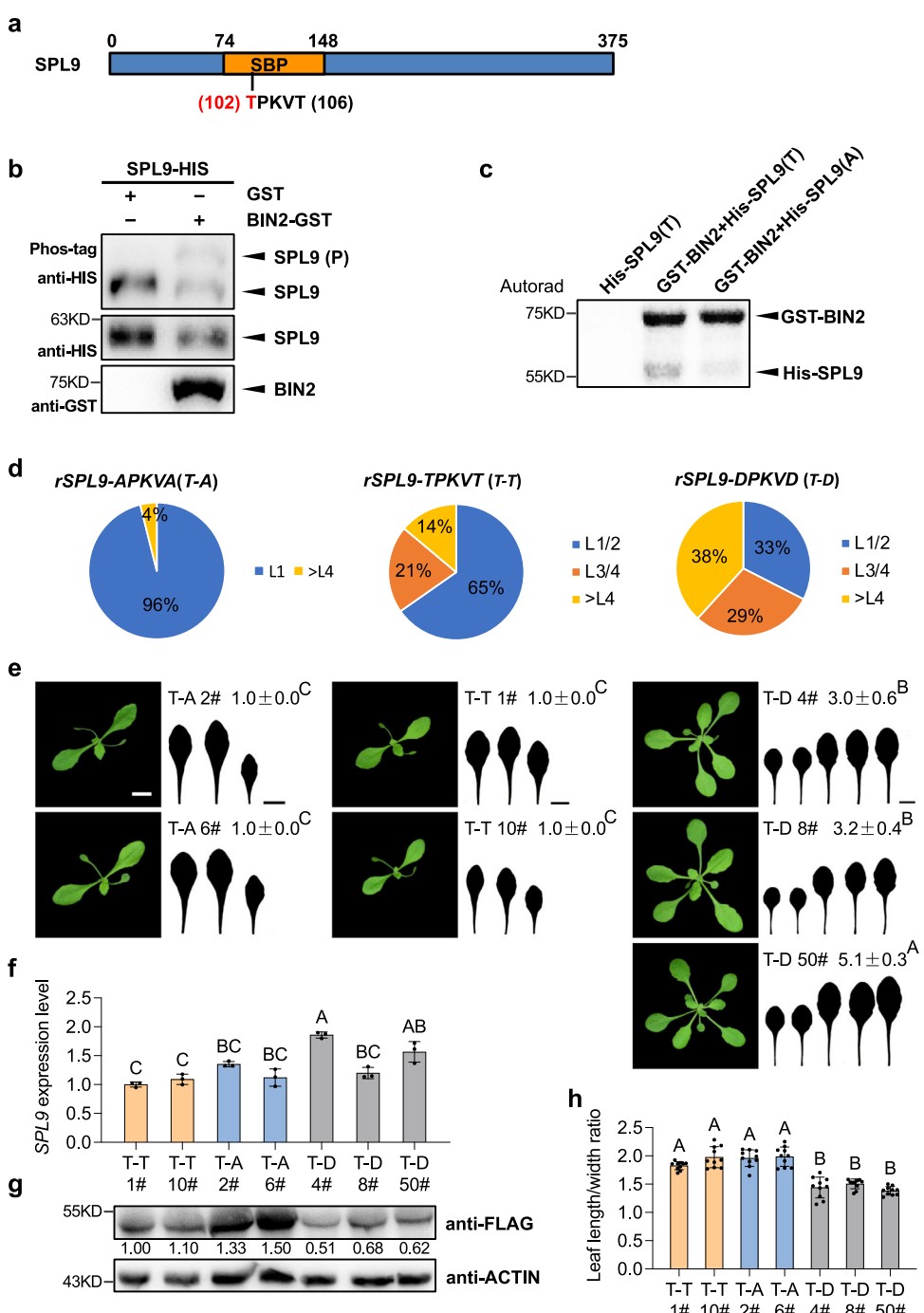

**Fig. 5 | BIN2 phosphorylates SPL9 and the phosphorylation site in SPL9 is required for its normal function to promote vegetative phase change. a** A schematic diagram of the SPL9 protein. The TPKVT motif is the typical conserved phosphorylation recognition motif of the GSK3 kinase. The threonine in red indicates the phosphorylated site. **b** SPL9 is phosphorylated by BIN2 in vitro. SPL9-His was incubated with GST or BIN2-GST at 30 °C for 1 h. Protein was separated in a SDS-PAGE gel containing phos-tag regent and detected by anti-GST and anti-His antibodies. SPL9 (P), phosphorylated SPL9 with slower gel mobility shift. **c** In vitro kinase assay. Kinase assays were performed with purified GST-BIN2 and His-SPL9 (T) or mutant His-SPL9 (A). **d** Pie charts show phenotypic distribution of the first leaf with abaxial trichomes in the primary *rSPL9-APKVA* (*T-A*), *rSPL9-TPKVT* (*T-T*), and *rSPL9-DPKVD* (*T-D*) transgenic plants. About 70 primary independent transformants were characterized. **e** Phenotypic characterization of representative transgenic plants transformed with *T-A*, *T-T*, and *T-D* with comparable levels of the SPL9 transcript. 21-day-old T-T 1#, 10#, T-A 2#, 6#, T-D 4#, 8#, and 50# lines grown

in short days were used for phenotypic analysis. Numbers indicate the first leaf with abaxial trichomes ($n = 20$ plants, ±SD). Different letters indicate significant difference between genotypes using one-way ANOVA at $P < 0.001$. Scale bar = 1 cm. **f**, **g** SPL9 transcript level (**f**) and protein level (**g**) in 14-day-old T-T 1#, 10#, T-A 2#, 6#, T-D 4#, 8#, and 50# transgenic lines in short days. Data are means ± SD from a representative experiment with three technical replicates for each sample. Different letters indicate significant difference using one-way ANOVA at $P < 0.01$. All experiments were repeated 3 times biologically. Numbers between two blots indicate the relative normalized value for each sample. The intensity of each sample was first normalized to its corresponding ACTIN, then the resultant value was normalized again to the value of T-T 1#. The band intensity was determined using image J. **h** Leaf length/width ratio of the first leaf from 40-day-old T-T 1#, 10#, T-A 2#, 6#, T-D 4#, 8#, and 50# plants. Different letters indicate significant difference between genotypes using one-way ANOVA at $P < 0.001$ ($n = 10$ plants, ±SD).

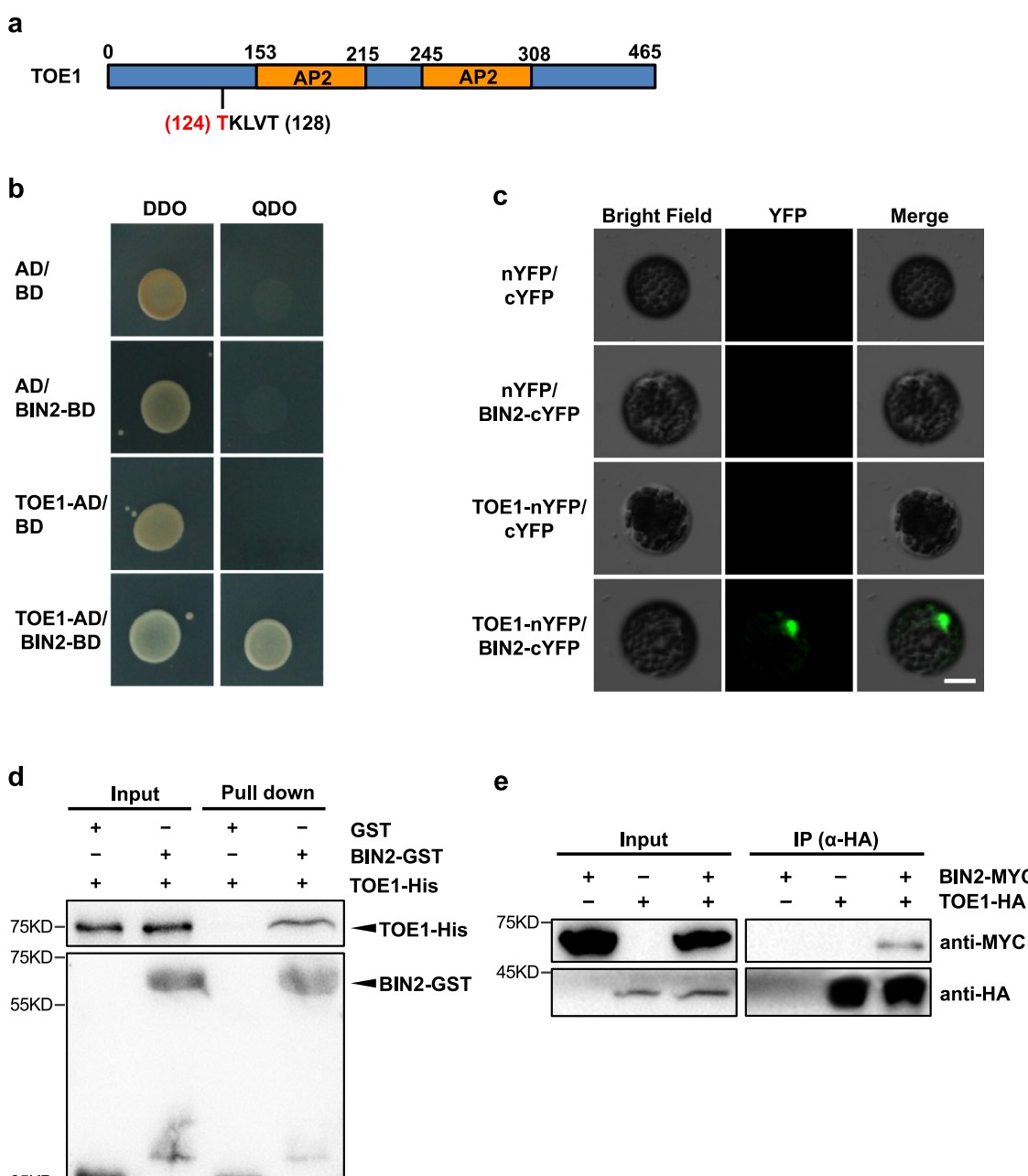

**Fig. 6 | BIN2 physically interacts with TOE1. a** A schematic diagram of the TOE1 protein. The TKLVT motif is the classical phosphorylation motif of BIN2. The threonine in red indicates the phosphorylated site. **b** BIN2 interacts with TOE1 in yeast. BIN2 was fused to pGAD (AD), and TOE1 was fused to pGBK (BD). AD and BD empty vectors were used as the negative controls. **c** BIN2 interacts with TOE1 in vivo in the nucleus as shown by BiFC assay. BIN2 was fused to cYFP, and TOE1 was fused to nYFP. The recombinant plasmids were transformed into the WT Arabidopsis protoplast. Scale bar = 10 μm. **d** BIN2 interacts with TOE1 in vitro in a pull-down assay. TOE1-His was incubated with GST or GST-BIN2, and proteins immunopreci-pitated with glutathione-agarose beads were detected with an anti-His and an anti-GST antibody, respectively. **e** BIN2 interacts with TOE1 in vivo in an CoIP assay. Total protein was extracted from Arabidopsis protoplasts containing *35 S::BIN2*-MYC, *35 S::3×HA-rTOE1*, and the *35 S::BIN2-MYC/35 S::3×HA-rTOE1* combination, and incu-bated with beads conjugated with an anti-HA. Coimmunoprecipitated proteins were detected with an anti-MYC and an anti-HA antibody, respectively. All experi-ments were repeated 3 times biologically.

WT protoplast. YFP fluorescence was observed in the transformed cell nuclei when BIN2-cYFP and TOE1-nYFP were co-transformed but not in all other negative control samples (Fig. 6c). This result demonstrated that BIN2 and TOE1 interact with each other in vivo in Arabidopsis cells. We also performed a protein pull-down assay to test if BIN2 interacts with TOE1 in vitro. We purified GST, GST-BIN2, and 6×His-TOE1 pro-teins from *E. coli*, and incubated the GST and GST-BIN2 proteins with His-TOE1 in vitro separately. Western blotting using an anti-His and an anti-GST antibody indicated that BIN2 also interacts with TOE1 in vitro

(Fig. 6d). To further confirm if BIN2 and TOE1 interact with each other *in planta*, we performed a CoIP assay. We co-expressed TOE1-HA and BIN2-MYC fusion proteins in the protoplast isolated from Arabidopsis leaves, and incubated with agarose beads conjugated with the anti-HA antibody, then detected the eluate with an anti-MYC and an anti-HA antibody, respectively. Western blotting result indicated that TOE1 and BIN2 were coimmunoprecipitated (Fig. 6e). Taken together, our results demonstrate that BIN2 physically interacts with TOE1 both in vitro and in vivo.

## Phosphorylation of TOE1 by BIN2 is critical for its function in vegetative phase change

To investigate the role of BIN2-mediated phosphorylation of TOE1, we again performed an LC-MS/MS analysis by transforming the *UBI10::3×FLAG-rTOE1* construct insensitive to miR172 into the protoplast from Ws and *bin2-3bil1bil2*, respectively. LC-MS/MS indicated that the 124[th] threonine in TOE1 was phosphorylated in WT (Fig. 6a, Supplementary Fig. 4a). To further confirm that BIN2 phosphorylates TOE1 in vitro, we performed an in vitro kinase assay. GST-BIN2, 6×His-TOE1 (T) protein in which the 124[th] amino acid is the wild-type threonine, and 6×His-TOE1 (A) in which the 124[th] threonine was replaced with an alanine were purified from bacterium. When GST-BIN2 was incubated with 6×His-TOE1(T) and radiolabeled ATP, autoradiographs showed two bands with the top band corresponding to the auto-phosphorylated GST-BIN2 and the bottom band corresponding to the phosphorylated His-TOE1 (Fig. 7a). However, when His-TOE1 (A) was incubated with GST-BIN2, the intensity of the bottom band corresponding to His-TOE1 in the autoradiograph is greatly reduced (Fig. 7a). This result further confirms that TOE1 is a substrate of BIN2, and the 124[th] threonine in the TOE1 protein is critical for its phosphorylation by BIN2.

To investigate the function of TOE1 phosphorylation by BIN2 in vegetative phase change, we mutated the 124[th] threonine in the TKLVT motif to AKLVT, and to DKLVT, respectively. Then we generated constructs expressing the wild-type form of *UBI10::3×FLAG-rTOE1* (*rTOE1-TKLVT*) insensitive to miR172 regulation, and mutated forms of *UBI10::3×FLAG-rTOE1* (*rTOE1-AKLVT*) and *UBI10::3×FLAG-rTOE1* (*rTOE1-DKLVT*) under the regulation of the Arabidopsis *Ubiquitin10* promoter, and transformed these constructs into Col-0. We first characterized the vegetative phase change phenotype of 20 different primary transgenic lines transformed with each construct in short days. About 70% of *rTOE1-AKLVT*, 40% of *rTOE1-TKLVT*, and 0% of *rTOE1-DKLVT* plants produced abaxial trichomes later than leaf nine (>L9), respectively (Fig. 7b), suggesting that constitutive phosphorylation of TOE1 impaired its function significantly to repress vegetative phase change. To see how phosphorylation affects TOE1 function and its stability, we first generated homozygous lines with a single T-DNA insertion in progenies from different transgenic lines. Next, we determined the transcript level of *TOE1* in those transgenic lines, and compared the phenotype of different transgenic lines with comparable levels of TOE1 transcript (Fig. 7c, d). Consistent with the above phenotypic characterization result, transgenic lines with *rTOE1-AKLVT* (T-A) produced abaxial trichomes on about leaf 9.7, transgenic lines with *rTOE1-TKLVT* (T-T) produced an intermediate vegetative phase change phenotype with abaxial trichomes on about leaf 9.1, whereas transgenic lines with *rTOE1-DKLVT* (T-D) had the weakest phenotype with abaxial trichomes on about leaf 7.7 (Fig. 7c). To see how the observed phenotype links to the TOE1 protein level and how phosphorylation affects the level of TOE1, we performed Western blotting using an anti-FLAG antibody to determine the level of the TOE1 protein in these transgenic lines. Among different transgenic lines, T-A 9#, 15#, and 16# accumulated relatively more TOE1 protein, T-T 8#, and 15# accumulated an intermediate level of the TOE1 protein, whereas T-D 1#, 8#, 10#, and 11# accumulated the lowest level of the TOE1 protein (Fig. 7e). These results imply that BIN2-mediated phosphorylation of the *TKLVT* motif is critical for TOE1 function to delay vegetative phase change by destabilizing the TOE1 protein and sequestering its function.

## Coordinated regulation of TOE1 at the protein level regulates vegetative phase change

Our results above suggest that BIN2 functions in a dual manner to regulate vegetative phase change with opposite outcomes for TOE1: one is to interact with SPL9 physically to destabilize it, thus reducing the level of miR172 to increase the TOE1 protein level, the other is to interact with TOE1 downstream of SPL9 to destabilize it directly. To answer the question of how these two modes of regulation by BIN2

contributes to TOE1 protein accumulation and vegetative phase change, we determined the level of TOE1 in the protoplast from *dwf5* mutant transformed with *UBI10::3×FLAG-rTOE1* insensitive to miR172 and *UBI10::3×FLAG-sTOE1* sensitive to miR172,respectively. The purpose of using these two constructs is to understand how the absence and presence of miR172 contribute to TOE1 accumulation. In *dwf5* transformed with *UBI10::3×FLAG-rTOE1* insensitive to miR172 regulation, the level of TOE1 was significantly reduced by about two-fold compared with WT (Fig. 8a), this is consistent with the expectation that the elevated level of BIN2 kinase activity in *dwf5* (Supplementary Fig. 1a) would accelerate the degradation of TOE1. This also implies that the BIN2-TOE1 regulatory mode is important to keep TOE1 from overaccumulation. However, in *dwf5* transformed with *UBI10::3×FLAG-sTOE1* sensitive to miR172 regulation, the TOE1 protein was elevated about 2.2-fold in *dwf5* compared with WT(Fig. 8b). This is well in agreement with the delayed vegetive phase change phenotype of *dwf5*, and the genetic result that *toe1 toe2* is completely epistatic to *dwf5* with respect to the production of abaxial trichomes (Fig. 3a). These results suggest that PTM of the TOE1 protein by both miR172 and BIN2 is important to maintain a homeostatic level of TOE1 in order for normal vegetative phase change to take place, and the regulation of TOE1 through the SPL9-miR172 pathway plays a predominant role in contrast to the regulation by BIN2.

## Discussion

Vegetative phase change constitutes a critical phase in plant development, and it is regulated by both endogenous and exogenous cues. Endogenously, it is regulated by the conserved sequential action between miR156 and miR172[5]. miR156 acts to repress the expression of ten different *SPL* genes posttranscriptionally to coordinate different aspects of vegetative trait development, especially leaf shape and leaf epidermal trait development. miR159 and some epigenetic modifiers act upstream of miR156 to regulate vegetative phase change[43–45]. Exogenous cues, such as sugar, phosphorus, temperature, photoperiod, phytohormone, affect vegetative phase change by modulating the expression of genes in the miR156-SPL-miR172 pathway[16–19,21,46–50]. However, the mechanism of how these exogenous factors is involved in vegetative phase change remains to be elucidated.

In plants, GA promotes, while JA delays, vegetative phase change in *Arabidopsis*, maize, and rice[2,16–18]. Only the mechanism of how GA promotes vegetative phase change has been partially elucidated. In this study, we showed that another phytohormone, BRs, also plays an important role in promoting vegetative phase change in *Arabidopsis*. This is the fourth phytohormone so far shown to regulate vegetative phase change. Plants defective in BR biosynthesis exhibited a delayed vegetative phase change phenotype. This delayed phenotype is mainly attributable to the reduction in the abundance of miR172 as a result of the proteolytic degradation of SPL9,and to a corresponding increase in the level of the TOE1 protein due to the dual negative regulation by both miR172 and BIN2 in *dwf5*. This result is in well agreement with a previous report that overexpression of miR172 could suppress the defect of *bak1*[51]. Based on our results from this study, we proposed a model for coordinated regulation of vegetative phase change by BRs and the age pathway (Fig. 8c). In WT, the presence of normal levels of BRs suppresses the activity of BIN2 to phosphorylate SPL9 and TOE1, thereby maintaining a normal level of SPL9 and TOE1 for plants to proceed to vegetative phase change; in *dwf5*, the reduced levels of BRs lead to an elevated level of BIN2 (Supplementary Fig. 1a), thus increasing the phosphorylated form of SPL9 and TOE1 simultaneously for subsequent proteolytic degradation with the regulation of TOE1 by SPL9-miR172-TOE1 outweighing the BIN2-TOE1 mode. As a result, TOE1 accumulates to a relatively higher level in *dwf5* than that in WT to delay vegetative phase change. Therefore, BRs participate in vegetative phase change by PTM of the SPL9 and TOE1 proteins simultaneously via BIN2, and this is achieved by the physical interaction between BIN2

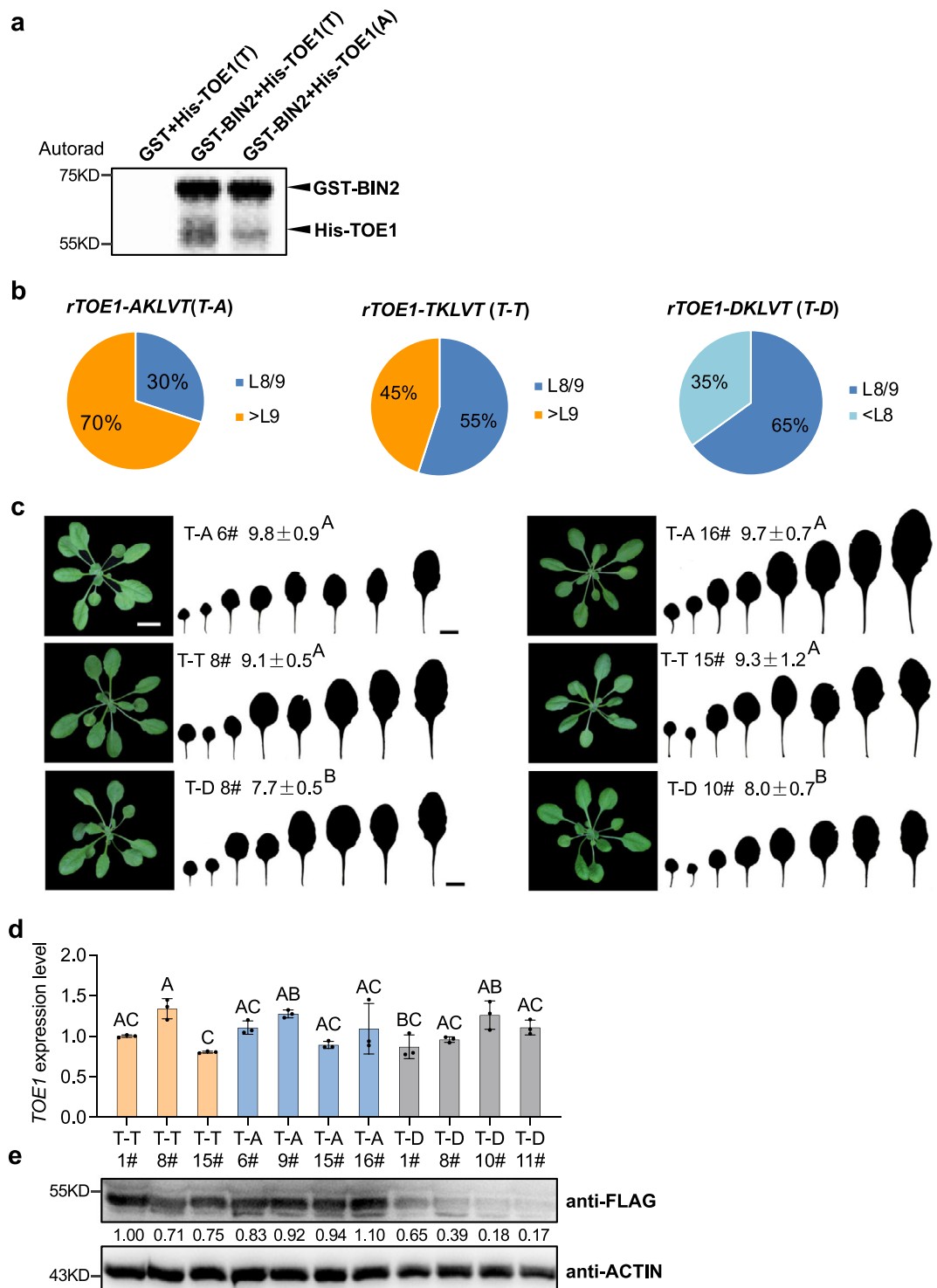

**Fig. 7 | BIN2 phosphorylates TOE1 and the phosphorylation site in TOE1 is required for its normal function to delay vegetative phase change. a** Kinase assays for purified GST-BIN2 and His-TOE1 (T) or His-TOE1 (A) proteins. **b** Pie charts show phenotypic distribution of the first leaf with abaxial trichomes in the primary *rTOE1-AKLVT* (*T-A*), *rTOE1-TKLVT* (*T-T*), and *rTOE1-DPVLT* (*T-D*) transgenic plants. About 20 primary independent transformants were characterized. **c** Phenotypic characterization of representative transgenic plants transformed with *rTOE1-AKLVT* (*T-A*), *rTOE1-TKLVT* (*T-T*), and *rTOE1-DPVLT* (*T-D*) with comparable levels of the TOE1 transcript. 21-day-old T-T 8#, 15#, T-A 6#, 16#, T-D 8#, and 10# lines grown in short days were used for phenotypic analysis. Numbers indicate the first leaf with abaxial trichomes (*n* = 15 plants, ±SD). Different letters indicate significant

difference between genotypes using one-way ANOVA at *P* < 0.01. Scale bar = 1 cm. **d, e** TOE1 transcript level (**d**) and protein level (**e**) in 14-day-old T-T 1#, 8#, 15#, T-A 6#, 9#, 15#, 16#, T-D 1#, 8#, 10#, and 11# transgenic lines in short days. Data are means ± SD from a representative experiment with three technical replicates for each sample. Different letters indicate significant difference using one-way ANOVA at *P* < 0.01. Numbers between two blots indicate the relative normalized value for each sample. The intensity of each sample was first normalized to its corresponding ACTIN, then the resultant value was normalized again to the value of T-T 1#. The band intensity was determined using image J. All experiments were repeated 3 times biologically.

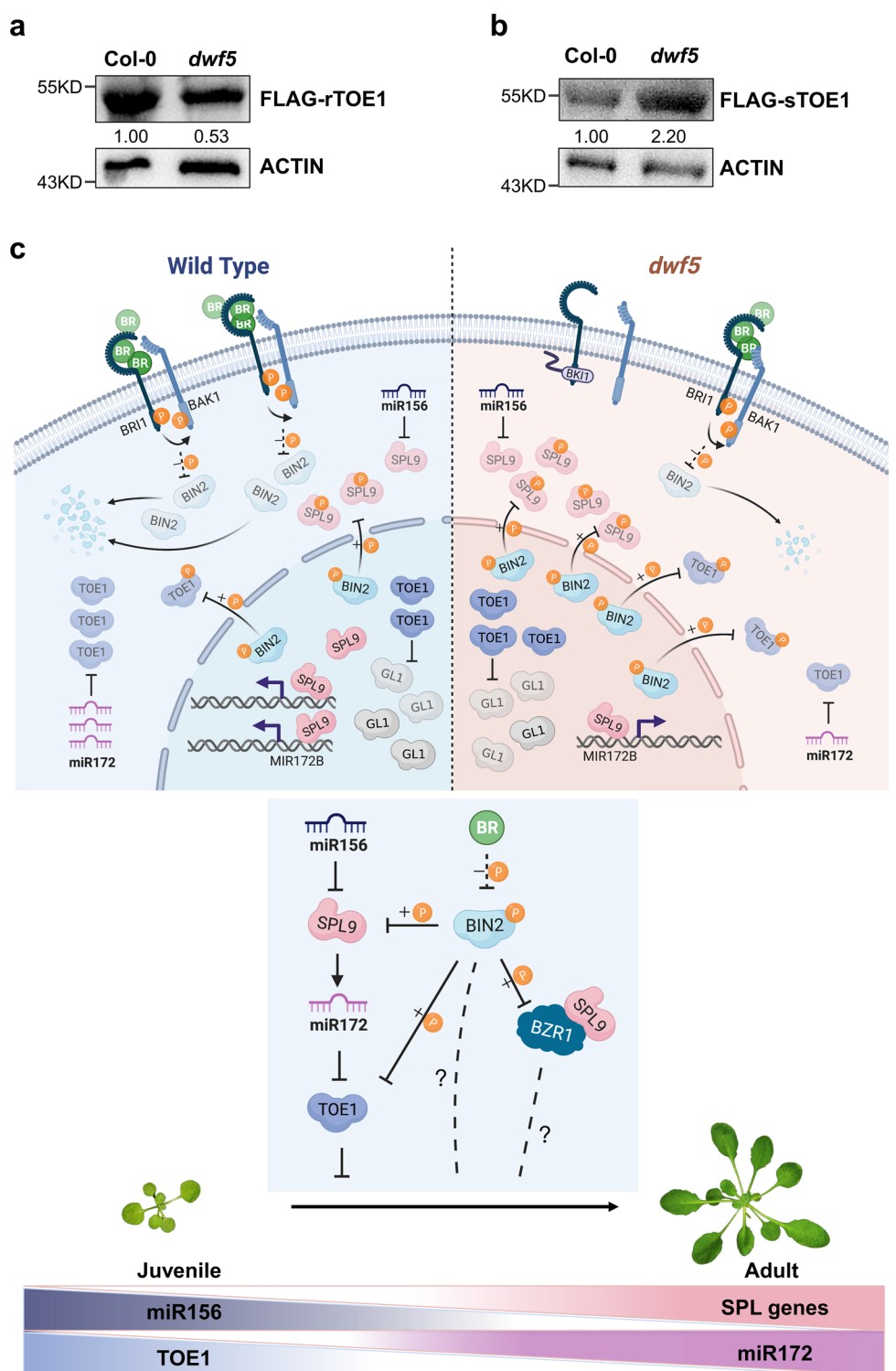

and SPL9, as well as the interaction between BIN2 and TOE1. It is of interest to see that BIN2 functions in a dual manner to post-transcriptionally modify SPL9 and TOE1 in the same genetic pathway with opposite outcomes for TOE1. During vegetative phase change, miR156 expression declines, while the expression of *SPL* genes increases[4,5]; likewise, miR172 expression increases, while *TOE1* expression declines[52] (Fig. 8c). A high level of miR156 at the juvenile phase renders a low level of SPL9 to alleviate the repression of TOE1 by miR172, leading to over-accumulation of TOE1. Therefore, it is possible, in this scenario, that the BIN2-TOE1 mode is initiated to safeguard a homeostatic level of TOE1 required for normal juvenile development.

This also can be inferred from the result that the miR172-sensitive form of TOE1 was reduced, while the miR172-insensitive form of TOE1 was elevated when the GSK3 kinase activity was abolished in *bin2-3bil1bil2* (Supplementary Fig. 4b, c).

A previous study indicated that S163 in IPA1 alters the DNA binding specificity in response to the infection by fungus to sustain a balance between growth and immunity[14]. However, how this pattern of phosphorylation is regulated by upstream factors remains unknown. In this study, we identified two GSK3 kinase classical phosphorylation motifs in both SPL9 and TOE1: one is T-P-K-V-T in the conserved SBP domain of SPL9, the other is T-K-L-V-T in TOE1. The status of

**Fig. 8 | Coordinated regulation of TOE1 posttranscriptionally by the age pathway and BRs during vegetative phase change. a, b** The FLAG-rTOE1 protein insensitive to miR172 level in *dwf5* (**a**), the FLAG-sTOE1protein sensitive to miR172 level in *dwf5* (**b**). *Ubi10::3×FLAG-TOE1* or *Ubi10::3×FLAG-rTOE1* were transformed into the protoplasts from Col-0 and *dwf5*, respectively. Total protein was extracted from transformed protoplasts and detected by Western blotting using an anti-FLAG and anti-ACTIN antibody, respectively. Numbers between two blots indicate the relative normalized value for each sample. The intensity of each sample was first normalized to its corresponding ACTIN, then the resultant value was normalized again to the value of Col-0. The band intensity was determined using image J. All experiments were repeated 3 times biologically with similar results. **c** A model for BR-mediated vegetative phase change in *Arabidopsis*. DWF5 is the rate-limiting enzyme in the BR biosynthetic pathway. In WT, the presence of normal levels of BRs suppresses the activity of BIN2 to phosphorylate SPL9 and TOE1, thereby maintaining a normal level of SPL9 and TOE1 for plants to proceed to vegetative phase change; in *dwf5* with a reduced level of BRs, more phosphorylated BIN2 is accumulated and activated to destabilize more SPL9 and TOE1 by physical interaction. This leads to the downregulation of miR172 in *dwf5*. As the action of the SPL9-

miR172-TOE1 mode predominates over that of BIN2-TOE1, TOE1 is upregulated in *dwf5*, thereby reducing the expression of *GL1* to delay trichome production. Therefore, *dwf5* exhibited a delayed vegetative phase change phenotype. During vegetative phase change, miR156 expression declines, while the expression of *SPL* genes increases; likewise, miR172 expression increases, while *TOE1* expression declines. Due to the high expression of miR156 at the juvenile phase, very low expression of SPL9 alleviates the repression of TOE1 by miR172, leading to overaccumulation of TOE1. In this scenario, the BIN2-TOE1 mode is initiated to safeguard a homeostatic level of TOE1 required for normal juvenile development. Therefore, BIN2 functions in a dual manner to posttranscriptionally modify SPL9 and TOE1 with opposite outcomes for TOE1. The gradient change in the color of miR156, *SPL9*, miR172, and *TOE1* represents the gradual change in the abundance of these genes during vegetative phase change. In addition to its function to regulate SPL9 and TOE1, BIN2 also regulates vegetative phase change through affecting the interaction between BZR1 and SPL9, or other unknown pathways. **c** was created with BioRender.com and Microsoft Powerpoint. P phosphorylation. Molecules in gray color represents the degraded proteins.

phosphorylation mediated by BIN2 is required for the stability and function of SPL9 and TOE1 to regulate vegetative phase change. However, the fact that 33% of *rSPL9-DPKVD* plants, in which this motif is in the constant phosphorylated status, still produced abaxial trichomes on leaf 1 or 2 (Fig. 5d) suggests that there might exist other phosphorylation sites important for SPL9 stabilization and function. This assumption is in line with previous studies showing that BIN2 can also phosphorylate non-canonical serine and threonine sites, and those phosphorylation sites exist in the SPL9 protein[15,39].

The genetic interaction analysis between *dwf5* and genes in the miR156-SPL pathway indicated that miR172 and *toe1/toe2* are epistatic to *dwf5*, and *SPL9* and *DWF5* act synergistically with respect to the production of abaxial trichomes. However, the leaf shape of *Ubi10::172B dwf5*, *spl9-4 dwf5*, and *toe1 toe2 dwf5* resembled more those of *dwf5*, suggesting that *dwf5* is epistatic to *Ubi10::172B*, *spl9-4*, and *toe1 toe2* with respect to leaf shape. This genetic result also implies that BRs involve vegetative phase change by other yet-to-known factors other than SPL9 and TOE1. Those targets could be different *SPL* genes that share functional redundancy with *SPL9*, or could be other genes that play a role in leaf development.

One obvious challenge for our model in the current study is that if the delayed vegetative phase change phenotype of *dwf5* is attributable to the elevated level of BIN2, we would assume that BIN2 loss-of-function mutants exhibit a precocious vegetative phase change phenotype; however, plants triply mutant for the *GSK3*-like kinase genes, *bin2-3 bil1 bil2*, exhibited a delayed vegetative phase change phenotype in contrast to its wild-type Ws (Supplementary Fig. 3a). This is contradictory to our model. It was shown that the *bin2-3 bil1 bil2* triple mutant accumulates more phosphorylated form of BIN2 substrate, and there might exist additional GSK3-like kinases that function redundantly with BIN2[53]. If this is the case, we would anticipate that more SPL proteins are phosphorylated, destabilized and/or sequestered functionally to cause a delayed vegetative phase change phenotype. However, LC-MS/MS analysis revealed that SPL9 was not phosphorylated in *bin2-3bil1bil2*, instead SPL9 was elevated in *bin2-3bil1bil2* (Supplementary Fig. 2a, b), this result negated our assumption. Another possibility is that the GSK3 kinase family regulates vegetative phase change mainly by posttranscriptional modulation of TOE1. However, LC-MS/MS analysis demonstrated that TOE1 is only phosphorylated in Ws, but not in *bin2-3bil1bil2* (Supplementary Fig. 4a). As expected, the level of miR172-insensitive form of TOE1 was elevated by about 2-fold in *bin2-3bil1bil2*, whereas the miR172-sensitive form of TOE1 is slightly reduced (Supplementary Fig. 4b, c). We can anticipate that a reduced level of TOE1 would yield a precocious vegetative phase change phenotype; however, vegetative phase change was delayed in *bin2-3bil1bil2*. Therefore, the delayed vegetative phase change

phenotype of *bin2-3bil1bil2* could not be explained by its effect on the level of TOE1. Further analysis indicated that the level of *GL1*, one of the TOE1 targets and a positive regulator of leaf epidermal trichome development, was significantly downregulated in *bin2-3bil1bil2*, while other genes in the miR156-SPL-miR172 pathway remains comparable between Ws and *bin2-3bil1bil2* (Supplementary Fig. 3b). This result implies that the GSK3 kinase family also regulates vegetative phase change through affecting genes downstream of TOE1. Another challenge to our model is that we would expect that *bzr1-1D*, a dominant mutant of *BZR1*, should exhibit a precocious vegetative phase change phenotype; however, *bzr1-1D* also exhibited a delayed vegetative phase change phenotype rather than a precious one (Supplementary Fig. 5a); moreover, loss-of-function mutations in *BZR1* delayed vegetative phase change (Supplementary Fig. 5b, c). *BZR1* plays dual roles in the BR biosynthetic pathway by feedback inhibition of BR biosynthesis and by regulation of downstream growth responsive genes;[54,55] moreover, *BZR1* also has a BR-signaling independent regulatory role in anther development[56]. BZR1 was recently shown to physically interact with SPL9 to cooperatively regulate downstream genes[57]. Therefore, the complicated role of the GSK3 kinase family and *BZR1* in vegetative phase change still awaits further investigation.

In addition to their role in the regulation of vegetative phase change in *Arabidopsis*, BRs also have a complicated impact on leaf angle, plant height, and inflorescence architecture and other yield-related traits[23,58–62]. Therefore, our result of the interaction of the BR signaling pathway with SPL9 and TOE1 is also important from a practical point of view in regards to molecular breeding for agronomically important crops, especially for rice. *IPA1* promotes the establishment of ideal plant architecture[63,64], genetic modulation of the BR pathway in combination of *IPA1* may provide a way to breed for elite crops with a higher yield in the future.

## Methods
### Plant materials and growth conditions
All genetic stocks, except *bin2-3* and *bin2-3 bil1 bil2*, used in this study were in a Columbia-0 (Col-0) genetic background. spl9-4 (CS807258), *pSPL9::rSPL9*, *pSPL9::3×FLAG-rSPL9*, *pSPL9::GR-rSPL9*, *Ubi10::172B*, and *toe1-2 toe2-* were seed stocks as described previously[5,45]. *bzr1-1D* (CS65987) was obtained from the Arabidopsis Biological Resource Center (ABRC). *bin2-3* and *bin2-3 bil1 bil2* were kind gifts from Dr. Jianming Li, and they were in the Ws background. The *dwf5* mutant was backcrossed to WT five times before phenotypic characterization. Seeds were grown in a mixture of peat and vermiculite moss at a 1:1 ratio, and left at 4 °C for 2 days before transfer to the growth chamber. Plants were grown in 32-well flats under short-day conditions (10 h light and 14 h dark, 120 μmol/m²/s) at 22 °C.

## Phenotypic analysis

The plant age was measured from the time when seeds were transferred to the growth chamber. Abaxial trichomes were scored during 2–4 weeks after planting using a stereomicroscope. For leaf initiation rate, the number of leaves was observed and scored using a stereomicroscope every 4 days. For leaf shape analysis, fully expanded leaves were removed from plants, attached to cardboard with double-sided tape, flattened with transparent tape, and then scanned with a digital scanner. All of the experiments in this study were repeated three times.

## Map based cloning

To map the *dwf5* mutation, *dwf5* was first crossed to Ler. In the selfed F$_2$ population, plants with the same phenotype to *dwf5* were chosen for the mapping purpose. DNA from about 100 mutant plants was pooled, and PCR was carried out using makers across different five chromosomes to localize the mutation. Fine mapping was done using DNA from individual DNA to narrow down the mutation.

## Plasmid construction and generation of transgenic lines

To generate the *pDWF5::DWF5* construct, the genomic sequence of *DWF5*, containing the promoter, coding, and 3' sequence, was amplified from Col-0 using PCR, and cloned into the *Kpn*I site of *pCAMBIA1305.1*. This construct was then introduced into the Agrobacterium strain *GV3101* to transform the *dwf5* plants using a floral dipping method. For yeast two hybrid system, the *BIN2* ORF (open reading frame) was cloned into the *Eco*RI and *Bam*HI sites of the *GAL4* transcriptional activation domain to generate *pGAD-BIN2*, the *SPL9* ORF was cloned into the *Nde*I and *Eco*RI sites of the *GAL4* DNA-binding domain to generate the *pGBK-SPL9* construct, and the *TOE1* ORF was cloned into the *Nde*I and *Bam*HI sites of *pGBK* to generate the *pGBK-TOE1* construct. For Bimolecular Fluorescent Complimentary (BiFC) experiment, the *SPL9* ORF was inserted into the *Xba*I and *Sal*I sites of *pXY103* to generate *SPL9-nYFP*, the *BIN2* ORF was cloned into the *Bam*HI and *Sal*I sites of *pXY105* to generate the *BIN2-cYFP* construct, and *TOE1* ORF was cloned into the *Bam*HI and *Sal*I sites of *pXY103* to generate the *TOE1-nYFP* construct. To generate the *SPL9-HIS*, *TOE1-HIS* and *BIN2-GST* constructs for pull-down assay, the ORF sequences of *SPL9*, *TOE1* and *BIN2* were cloned into *pET-30a (+)* and *pPGH* (derived from *pGEX4T-2*), respectively. *pET-30a (+)-SPL9*, *pET-30a (+)-TOE1*, *pPGH*, and *pPGH-BIN2* were then introduced into the *Escherichia coli* strain Rosetta (DE3). For coimmunoprecipitation (Co-IP) analysis of *SPL9* and *BIN2*, the *SPL9* and *BIN2* ORF sequences were individually cloned into a modified *pCAMBIA1300* vector with a 3×FLAG or a GFP tag. For Co-IP analysis of *TOE1* and *BIN2*, the *TOE1* and *BIN2* ORF sequences were individually cloned into a modified *pBWA(V)HS* vector with an MYC or a HA tag. The resultant *35 S::3×FLAG-rSPL9*, *35 S::BIN2-GFP*, *35 S::BIN2-MYC* and *35 S::TOE1-HA* constructs were then introduced into the Agrobacterium strain GV3101 to infiltrate tobacco (*Nicotiana benthamiana*) leaves or were directly transformed into the Arabidopsis protoplasts. To introduce mutations into the conserved T-P-K-V-T motif in SPL9 and T-K-L-V-T motif in TOE1, overlapping PCR was carried out to amplify the full-length miR156-insensitive *SPL9* and miR172-insensitive *TOE1* coding sequence using primers with introduced mutations, the PCR product was then cloned into the *Bam*HI and *Nco*I sites in the *pSPL9::3×FLAG-rSPL9* and *UBI10::3×FLAG-rTOE1* construct[5]. These constructs were introduced into the Agrobacterium strain GV3101 to transform WT or were directly transformed into the Arabidopsis protoplasts. To generate the *pBIN2::eGFP-BIN2* construct, an about 2000-bp *BIN2* promoter sequence was first amplified by PCR, then the sequence was inserted into the *Kpn*I and *Nco*I sites in the *pCAMBIA1305.1* vector, and eGFP was fused to the N-terminus of the *BIN2* genomic sequence using overlapping PCR, the PCR fragment was finally cloned into the *Nco*I and *Pml*I sites of the vector harboring the *BIN2* promoter sequence. This plasmid was introduced into the

Agrobacterium strain GV3101 to transform WT. All primers used for cloning are listed in Supplementary Table.

## BL addition experiment

Seeds were germinated on 1/2 MS plate with 0 or 10 nM BL (Sigma) in the short-day growth chamber. 10-day-old seedlings were then transferred to the 1/2 Hogland liquid medium supplemented with 0 or 10 nM BL. The Hogland medium was replaced with new one every 3 days until for abaxial trichome and leaf shape analysis.

## Gene expression analyses

For gene expression analyses, 12-day-old seedlings without cotyledons were collected and stored at −80 °C until use. Tissues were ground into fine powder in liquid nitrogen using a homogenizer. Total RNA was extracted using TRIzol (Ambion) and digested with RNase-free DNase (TaKaRa) according to the manufacturer's protocol. Digested RNA was quantified, then reverse transcribed (RT) was done with a PrimerScript II 1st Strand cDNA Synthesis Kit (TaKaRa). Oligo-dT primer and miRNA-specific primers were used for preparing the first-strand cDNA of mRNA and miRNA, respectively. Real-time PCR was performed using diluted cDNA on Step One Plus (ABI) real-time PCR machine. *TUBLIN2* and *AtSnoR101* were served as the internal controls for mRNAs and miRNAs analyses, respectively. All qRT-PCR primers are listed in Supplementary Table.

## Yeast two-hybrid

The bait plasmid *pGBK-BIN2* and the prey plasmid *pGAD-SPL9*, *pGAD-TOE1* were co-transformed into yeast strain AH109. The transformants were screened on the SD/-Leu/-Trp (Double Dropout Supplement, DDO) medium by following the procedure as described in Yeast Protocols Handbook (Clontch, PT3024-1). Healthy colonies from DDO medium were chosen and inoculated onto the SD/-Leu/-Trp/-Ade/-His (Quadruple Dropout Supplement, QDO) medium for further analysis.

## BiFC assay

The recombinant *SPL9-nYFP*, *TOE1-nYFP* and *BIN2-cYFP* plasmids were cotransformed into the WT protoplast isolated using a Tape-*Arabidopsis* Sandwich method through the PEG-mediated method[64]. Leaves were collected from 3- to 4-week-old plants grown in SD condition. The lower epidermal surface was first affixed to a strip of Magic tape (3 M, Scotch), then the leaves were pulled away with the Time tape (Time Med) affixed to the upper epidermal surface. The peeled leaves with the Time tape were transferred to the enzyme solution [1.5% (w/v) cellulase R10 (Yakult), 0.4% (w/v) macerozyme R10 (Yakult), 0.4 M mannitol, 10 mM CaCl$_2$, 20 mM KCl, 0.1% BSA and 20 mM MES pH 5.7], and were digested at room temperature for 1–2 h. The digested solution then was filtered through a 75-μm nylon mesh. The protoplasts were centrifuged at 100 × *g* for 2 min and washed with the W5 buffer (154 mM NaCl, 125 mM CaCl$_2$, 5 mM KCl, 5 mM glucose, and 2 mM MES, pH 5.7) twice, and then incubated on ice for 30 min. For transformation, the protoplasts were centrifuged and resuspended in ice-cold MMG buffer (0.4 M mannitol, 15 mM MgCl$_2$, and 4 mM MES, pH 5.7). 100 μL protoplasts were mixed with 10 μg plasmid each and 120 μL PEG-Ca$^{2+}$ buffer [40% (w/v) PEG400, 0.2 M mannitol, 100 mM CaCl$_2$]. The mixture was incubated at room temperature for 10 min, and washed twice with the W5 buffer. Transformed protoplasts were resuspended in 1 mL W5 buffer and transferred into 6-well plates and incubated for 12–16 h. Protoplasts were finally stained with hoechst33342 (Sigma) for 0.5 h, and were observed with a laser scanning confocal microscope (Zeiss, LSM510).

## In vitro pull-down assay

*E. coli* transformed with different expressing vectors was induced with 0.2 mM isopropylthio-β-galactoside (IPTG), and was incubated in a shaker at 16 °C for 20 h. Purification of His-tagged recombinant protein

was performed as described in the Ni-NTA Purification System (Novex by Life Technology), the GST-tagged protein was purified according to a protocol described previously[65]. For SPL9-His and TOE1-His, cells were collected and resuspended in 10 mL native purification buffer (50 mM $NaH_2PO_4$ pH8.0, 50 mM NaCl, 1 mM PMSF). The cell lysate was sonicated on ice with 7 s ON/7 s OFF at 30% intensity for 10 min and centrifuged at 12000 rpm for 20 min. The supernatant was transferred to a balanced column with Ni-NTA resin (TransGen) and incubated at 4 °C for 4 h with gentle agitation to keep the resin suspended in the lysate solution. The resin was settled by gravity and washed with 8 mL native wash buffer (50 mM $NaH_2PO_4$ pH8.0, 50 mM NaCl, 20 mM imidazole) 4 more times and finally was eluted with 1 mL native elution buffer (50 mM $NaH_2PO_4$ pH8.0, 50 mM NaCl, 300 mM imidazole). For BIN2-GST, a similar purification process was followed except for the buffer and resin used. Cells were resuspended with PBS-L [50 mM $NaH_2PO_4$ pH 8.0, 150 mM NaCl, 1 mM EDTA, 0.2% Triton X-100 (v/v), and 1 mM PMSF]. After binding with the lysate, the GST resin (TransGen) was washed with PBS-EW (50 mM $NaH_2PO_4$ pH 8.0, 150 mM NaCl, 1 mM EDTA, 1 mM DTT) and eluted with TNGT [50 mM Tris-HCl pH 8.0, 100 mM NaCl, 0.01% Triton X-100 (v/v), 30 mM Reduced glutathione, 1 mM DTT]. For the pull-down assay, the purified SPL9-His and TOE1-His recombinant proteins were incubated with resin bound with GST or GST-BIN2 at 4 °C for 4 h with gentle rotation. The beads were then collected and boiled in the SDS loading buffer for 10 min after washing five times with PBS-EW. Protein samples were separated in a 10% SDS-PAGE gel and were examined by immunoblotting using an anti-His (Sangon) and an anti-GST (TransGen) antibody, respectively.

## Co-IP assay

*Agrobacterium* harboring *35 S::3×FLAG-rSPL9*, *35 S::BIN2-GFP*, and *35 S::3×FLAG* constructs was incubated in LB medium containing 50 mg/mL Kanamycin, 50 mg/mL Rifampicin, 10 mM MES pH5.6, and 20 μM Acetosyringone. Cells at the log growth phase were collected and resuspended in the infiltration solution (10 mM MES pH5.6, 150 μM AS, 10 mM $MgCl_2$, 0.5% Glucose) to an OD value of 0.8. The Agrobacterium solution with *35 S::BIN2-GFP* was mixed with *35 S::3×FLAG* or *35 S::3×FLAG-SPL9* at a 1:1 ratio, respectively. The mixed solution was then used to infiltrate tobacco (*Nicotiana benthamiana*) leaves. 100 μg of each recombinant *35 S::BIN2-MYC* and *35 S::TOE1-HA* plasmids was co-transformed into the WT protoplast. Total protein was isolated from *Nicotiana benthamiana* leaves 2 days after infiltration or from transformed protoplast with the IP buffer [50 mM HEPES pH 7.5, 150 μM NaCl, 1 mM EDTA, 0.2% Tritron X-100, protease inhibitor cocktail (Sigma)], and was incubated with agarose beads conjugated with an anti-FLAG (Sigma) or an anti-HA (Sigma) antibody at 4 °C for 4 h with gentle agitation. The beads were collected and washed five times with the IP buffer and eluted with 3×FLAG peptide (Sigma). The immunoprecipitate was boiled for 10 min in the SDS loading buffer and detected by an anti-FLAG (Beyotime), an anti-GFP (Sangon), an anti-MYC (Sangon), and an anti-HA (Sangon) antibody, respectively.

## In vitro kinase assay

The purified His-SPL9 protein was incubated with GST or GST-BIN2 protein in 20 μl kinase buffer (25 mM Tris-HCl pH7.5, 10 mM $MgCl_2$, 0.5 mM DTT, 1 mM ATP) at 30 °C for 1 h, respectively. Reactions were stopped by adding the protein loading buffer, and were boiled for 10 min. The samples were separated on 8% SDS-PAGE gels with or without 25 μM Phos-tag reagent (NARD Institute, Ltd) and 100 μM $MnCl_2$ (Sigma), and the blot was then detected with an anti-His and an anti-GST antibody, respectively. For in vitro kinase assays, all recombinant proteins were purified from *Escherichia coli Rosetta*. The reaction was performed in 20 mL kinase buffer (20 mM Tris, pH 7.5, 100 mM NaCl, 12 mM $MgCl_2$, 10 μM ATP, and 10 μCi of [γ-$^{32}$P] ATP). The kinase assays were performed at 37 °C for 30 min, and then were incubated with SDS loading buffer at 100 °C for 10 min. The samples

were separated in 12% SDS-PAGE gels and then phosphorylation was detected in the dried gels exposed to phosphor screens. The autoradiograph (Autorad) signal was detected by a Typhoon9410 phosphor imager.

## Liquid chromatography-mass spectrometry (LC-MS/MS) analysis

For the analysis of the phosphorylation site in SPL9 and TOE1 by BIN2, the SPL9-FLAG and TOE1-FLAG fusion proteins were purified from Arabidopsis protoplasts. The recombinant *pSPL9::3xFLAG-rSPL9* or *Ub10::3xFLAG-rTOE1* plasmid was transformed into Ws and *bin2-3bil1bil2* protoplasts, respectively. Total protein was isolated from protoplasts with the IP buffer, and then was incubated with agarose beads conjugated with an anti-FLAG (Sigma) antibody at 4 °C for 4 h with gentle agitation. The beads were collected and washed five times with the IP buffer and boiled for 10 min in the SDS loading buffer. Samples were separated in SDS-PAGE gels, then stained with Coomassie brilliant blue. The band of target protein with the correct size was cut with a scalpel for MS analysis. The samples in the gel were treated as described previously[66]. The dried polypeptide samples were first re-dissolved in Nano-HPLC Buffer A (0.1% formic acid-aqueous solution), and separated using the nano-HPLC liquid phase system Easy-NLC1200. The samples were then loaded to an automatic sampler and adsorbed to a Trap column (RP-C18, Thermo Inc.), and separated with an Analysis column (rp-c18, thermo Inc.) at a flow rate of 300nL/min. The hydrolysates were separated by capillary HPLC and analyzed by Q-Exactive mass spectrometry (Thermo Scientific). The scanning range of parent ions was set at 300-1600 m/z, and the Data Dependent Acquisition (DDA) scanning mode was employed. The 20 strongest fragment profiles (MS2 Scan) were collected after each full scan. Fragmentation was performed using high-energy collision dissociation (HCD, high energy) with NCE energy of 28 and dynamic removal time of 25 s. The resolution of MS1 was 45000 at M/Z 200, the AGC target was 1E6, and the maximum injection time was 50 ms. The resolution of MS2 was 15000, the AGC target was 1E5, and the maximum injection time was 50 ms. The ProteomeDiscover 2.1 software was used for database search. MS/MS spectra were searched using MaxQuant against the uniprot-Arabidopsis thaliana (Mouse-ear cress) [3702] UP000006548 database. The parameters for database search were set as follows: Fixed modifications: Carbamidomethyl (C); Variable modification: Oxidation (M), Acetyl (Protein N-term), and Phospho (STY); Digestion: trypsin; First search peptide tolerance: 20 ppm; Main search peptide tolerance: 4.5 ppm; Max missed: 2.

## Reporting summary

Further information on research design is available in the Nature Portfolio Reporting Summary linked to this article.

## Data availability

Source data are provided with this paper.

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

## Acknowledgements

This work is supported by the National Natural Science Foundation of China (31970191, 31770209, 32172595), the Natural Science Foundation of Zhejiang province (LZ22C020003, LY21C150002), start-up funds from Zhejiang Agriculture and Forestry University (2012FR025) to G.W. and S.F.. We thank Dr. Qingfu Ye at Zhejiang University for providing equipment for the $^{32}$P experiment and analysis, and Dr. Jianming Li for kindly providing the *bin2-1*, *bin2-3* and *bin2-3 bil1 bil2* seeds.

## Author contributions

B.Z., S.F. and G.W. conceived of the study and wrote the manuscript. B.Z. and L.W. made vectors and did yeast two-hybrid analysis. B.Z. performed the map-based cloning, genetic analysis, qRT-PCR and data analysis. B.Z. and Q.L. carried out MG132 treatment, pull down assay and western blotting. B.Z., Q.L. and Y.S. carried out Co-IP and the kinase assay. B.Z., X.S., and H.L. performed protoplast isolation and transformation. L.N., T.S., X.D., J.H. and M.J. provided materials and assistance in the kinase assay and Co-IP. S.F. and G.W. oversaw the entire study. All authors read and approved the manuscript.

## Competing interests

The authors declare no competing interests.
