## [Peer Review File · Nature Communications]

Coordinated regulation of vegetative phase change by brassinosteroids and the age pathway in ArabidopsisReviewer #1 (Remarks to the Author):

In this study, the author showed that the *Arabidopsis* *dwf5* mutant exhibits a delayed vegetative phase change phenotype. *DWR5* encodes an enzyme ($\Delta 7$ -sterol reductase) essential for BR biosynthesis. Accordingly, they showed that the *dwf5* mutant has reduced level of BR. Moreover, the authors showed that *BIN2* directly interacts with and phosphorylates *SPL9*, causing proteolytic degradation of *SPL9*, and subsequently reduced level of *miR172* and delayed phase change. The results provided new insights into the molecular mechanisms of BR regulation of vegetative phase change. In general, the data provided are of high quality, and conclusions are well justified.

I have several comments for the authors to consider and improve the manuscript:

1. I wonder why the expression of several AP2-like target genes, like *TOE1* and *TOE2*, was not obviously affected in the *dwf5* mutant, whereas the expression levels of *MIR172B* and mature *miR172* are notably reduced in the *dwf5* mutant? (Fig 2a,b). The authors also showed that *toe1 toe2* was completely epistatic to *dwf5* with respect to abaxial trichome production (Fig3a). These results seem contradicting to each other? Please clarify.
2. Previous studies showed that *SPL9* and *SPL15* are functionally similar in regulating many developmental processes, such as branching and flowering time. I wonder if the authors have checked the phenotype of *spl9* and *spl15* single mutant, and *spl9spl15* double mutants, with respect to phase change;
3. I wondered what is the vegetative phase change phenotype of *pSPL9::3xFLAG-rSPL9* (in wild type and *dwf5* mutant background)? Are they comparable to the *pSPL9::rSPL9* transgenic plants (in WT and *dwf5* mutant background respectively) (Fig 3a-c)?
4. I wonder whether the protein levels of *SPL9* in the different transgenic lines of *rSPL9-APKVA* (T-0A), *rSPL9-TPKVT* (T-T) and *rSPL(-DPKVD)* (T-D) correlate with the vegetative phase change phenotype (Fig. 5)?
5. The authors mentioned that the *bin2-3 bil1 bil2* triple mutant also displays a delayed vegetative phase change phenotype, which is opposite to the expectation and the working model. To resolve the contradiction, the levels of phosphorylation and protein levels of *SPL9* need to be determined.

Reviewer #2 (Remarks to the Author):

In this manuscript, the authors examined the function of plant steroid hormone, Brassinosteroid (BR), in the regulation of juvenile-to-adult transition (JAT) in *Arabidopsis*. They provided genetic and molecular evidence that BR functions through *BIN2* kinase to regulate *SPL9*/*miR172B*/*TOE1*/*TOE2* to regulate juvenile to adult transition in a pathway:

BR—I BIN2—I SPL9 ->miR172B—I TOE1 TOE2—I Juvenile-to-Adult Transition (JAT).
(see attached)

The study is of potential interests to the field, but some of the studies regarding molecular mechanisms are not well established. The current model cannot explain several evident genetic results. I have some questions and suggestions for the authors to address and improve the manuscript.

1. Why BR treatment has no effect on JAT in WT (Fig 1g)?
2. In Fig 2b, *MIR172B* is decreased, why *TOE1* and *TOE2* are not increased as *MIR172B* targets?
3. The author showed *SPL9* is less stable in *dwf5* mutant in Fig 1c/e, which should be confirmed by cycloheximide (CHX) treatment experiments.

4. The accumulation of SPL9 by BL treatment shown in Fig 1f is minimal (1.18X) and should be done with a time course experiment, for example with 0.5, 1, 2, 4 and 8 hr BL treatment.

5. The epistasis analysis in Fig 3 is very important and well done, but the description is not clear. In principle, overexpression of SPL9, MIR172B and loss-of-function toe1 toe2 mutant all suppressed dwaf5 mutant phenotype in JAT, supporting the linear model I summarized above. Please clearly describe these nice genetic results.

6. The phosphorylation of SPL9 by BIN2 shown in Fig 5 should be more vigorously established. The kinase assay should be done with isotope for better sensitivities. The mutant SPL9 protein with two putative BIN2 site changed to Ala should be tested in kinase assay with BIN2.

7. The effect of SPL (T-A) and SPL9 (T-D) mutations should be examined for A and D forms in terms of protein stability (Fig 5g). The author should identify transgenic lines with similar transcript levels to examine the protein levels and stability (i.e. CHX treatment). It's likely that T-A mutation stabilizes and T-D mutation destabilize the SPL9 proteins.

8. The bin2 loss-of-function and bzc-1D gain-of-function mutants have opposite phenotypes to what can be predicated from the current model. The authors did not provide a viable explanation. One possibility is that TOE1 and TOE2 genes are induced by BR (RNA-seq data in Wang et al., Molecular Plant. 7: 1303-1315). So it's possible that BR affects JAT through two branched pathways with opposite outcomes likely under different conditions (one through BIN2-SPL9/MIR172B/TOE1/2 and another through BIN2-BES1/BZR1/TOE1/2, see attached ppt). This can explain the two mutant phenotypes here as well as answer questions I raised in points 1 and 2. I hope the authors can test this possibility by examining TOE1 and TOE2 gene expression in various mutants.

Feel free to contact me, Yanhai Yin if you need any clarifications.

Reviewer #2 Attachment on the following page

BES1/BZR1

BR—| BIN2—| SPL9 → miR172B—| TOE1 TOE2—| Juvenile-to-Adult Transition (JAT).

Dear Editors and Reviewers,

Thanks for your constructive comments on our manuscript entitled “Coordinated regulation of the timing of the juvenile-to-adult transition by brassinosteroids and the age pathway in *Arabidopsis*” (NCOMMS-21-03803) submitted to Nature Communications. We are really grateful that those comments are very constructive and beneficial for us to improve the overall quality of the manuscript. Due to the pandemic and lock-down of many cities in China from 2021 to 2023, we had great difficulties getting the radioactive ³²P and reagents for our experiments. We are sorry that it took us much longer time than normal to finish all required experiments and to address the point-by-point comments raised by both reviewers. Below is our response to the comments/questions raised by both reviewers.

REVIEWER COMMENTS

Reviewer #1 (Remarks to the Author):

In this study, the author showed that the *Arabidopsis* *dwf5* mutant exhibits a delayed vegetative phase change phenotype. *DWR5* encodes an enzyme ($\Delta 7$ -sterol reductase) essential for BR biosynthesis. Accordingly, they showed that the *dwf5* mutant has reduced level of BR. Moreover, the authors showed that BIN2 directly interacts with and phosphorylates SPL9, causing proteolytic degradation of SPL9, and subsequently reduced level of miR172 and delayed phase change. The results provided new insights into the molecular mechanisms of BR regulation of vegetative phase change. In general, the data provided are of high quality, and conclusions are well justified.

I have several comments for the authors to consider and improve the manuscript:

1. I wonder why the expression of several AP2-like target genes, like TOE1 and TOE2, was not obviously affected in the *dwf5* mutant, whereas the expression levels of MIR172B and mature miR172 are notably reduced in the *dwf5* mutant? (Fig 2a,b). The authors also showed that *toe1 toe2* was completely epistatic to *dwf5* with respect to abaxial trichome production (Fig3a). These results seem contradicting to each other? Please clarify.

Response: Thanks for this good question on the expression of AP2-like genes in *dwf5* raised by both reviewers. miR172 has been shown by Xuemei Chen's group in 2004 that it functions to regulate AP2-like gene expression predominantly by translational inhibition. (Chen Xuemei, A microRNA as a translational repressor of APETALA2 in *Arabidopsis* flower development. *Science*, 2004, 303: 2022-2025.). Therefore, although miR172 expression is reduced, the transcript levels of *TOE1/TOE2* are not affected significantly in *dwf5* (Fig.2b). Our further work showed that the miR172-sensitive TOE1 protein level is significantly upregulated (Fig.8b). This result is also in line with the results that miR172 is reduced in *dwf5*(Fig.2b), and *toe1 toe2* is

completely epistatic to *dwf5* with respect to abaxial trichome production (Fig.3a). We added the description of how *AP2*-like genes is regulated by miR172 to the main text to avoid any confusion.

2. Previous studies showed that SPL9 and SPL15 are functionally similar in regulating many developmental processes, such as branching and flowering time. I wonder if the authors have checked the phenotype of *spl9* and *spl15* single mutant, and *spl9 spl15* double mutants, with respect to phase change;

Response: *SPL9* and *SPL15* function redundantly in regulating many developmental processes. The work from Scott Poethig's Lab has shown that miR156-targeted SPLs have overlapping as well as distinct roles in vegetative phase change (Xu, et al. Developmental Functions of miR156-Regulated *SQUAMOSA PROMOTER BINDING PROTEIN-LIKE (SPL) Genes* in *Arabidopsis thaliana*. PLoS Genet, 2016, 12: e1006263.). Among them, *SPL2*, *SPL9*, *SPL10*, *SPL11*, *SPL13* and *SPL15* have a much more significant role in vegetative development. Specifically, *spl9-4*, *spl11-1* and *spl13-1* produced a small increase in the number of juvenile leaves in both long and short days, whereas *spl2-1*, *spl10-2*, and *spl15-1* had no obvious effect on juvenile leaf number. Plants mutant for more than one of these genes had much stronger phenotypes however. The strongest interaction is between *spl9* and *spl13*. In long days, *spl9/13* had 6–8 more juvenile leaves and 3–5 more rosette leaves than Col, and in short days it had 15–16 more juvenile leaves and 6–7 more rosette leaves than Col. The addition of *spl15* (*spl9/13/15*) produced a further delay in vegetative phase change and a larger increase in rosette leaf number. In long days, *spl9/13/15* produced 11–14 more juvenile leaves and 9–16 more rosette leaves than Col, and in short days it produced 16–24 more juvenile leaves and 6–17 more rosette leaves than Col. We also grew those genotypes in our own lab and characterized the phenotype again in short days. Our result also indicates that *SPL9*, *SPL13*, and *SPL15* function redundantly to promote vegetative phase change (Fig.1).

Fig.1. Phenotype of 27-day-old WT, *spl9-4* and *spl9/13/15* plants grown in short days. Numbers indicate the first leaf with abaxial trichomes (n=20 plants, ±SD). Different letters indicate significant difference between genotypes using one-way ANOVA at $P < 0.01$. Scale bar = 1 cm.

3. I wondered what is the vegetative phase change phenotype of *pSPL9::3xFLAG-rSPL9* (in wild type and *dwf5* mutant background)? Are they comparable to the *pSPL9::rSPL9* transgenic plants (in WT and *dwf5* mutant background respectively) (Fig 3a-c)?

Response: We compared the vegetative phase change phenotype of *pSPL9::3xFLAG-rSPL9* and *pSPL9::rSPL9* in short days. The result shows that there is no significant difference in the vegetative phase change phenotype of *pSPL9::3xFLAG-rSPL9* and *pSPL9::rSPL9* transgenic plants when they are either in the WT or in the *dwf5* mutant background, respectively (Fig.2). However, the same transgene exhibited slightly but significant difference in the phenotype between WT and *dwf5* because *dwf5* affects SPL9 function (Fig.2). Therefore, the fusion of the *3xFLAG* epitope tag does not affect the function of SPL9 protein to promote vegetative phase change. Moreover, these two lines have been used in a previous paper published by the correspondence author (Wu et al., The sequential action of miR156 and miR172 regulates developmental timing in *Arabidopsis*. Cell 138, 4: 750-759).

Fig.2. Phenotype of 33-day-old *pSPL9::3xFLAG-rSPL9* and *pSPL9::rSPL9* transgenic plants in the WT and *dwf5* mutant background grown in short days. Numbers indicate the first leaf with abaxial trichomes ($n \geq 25$ plants, \pm SD). Different letters indicate significant difference between genotypes using one-way ANOVA at $P < 0.01$. Scale bar = 1 cm.

4. I wonder whether the protein levels of SPL9 in the different transgenic lines of rSPL9-APKVA (T-0A), rSPL9-TPKVT (T-T) and rSPL9-DPKVD (T-D) correlate with the vegetative phase change phenotype (Fig. 5)?

Response: We appreciate this constructive comment raised by both reviewers. According to the comments raised by both reviewers, we first characterized the vegetative phase change phenotype of more than 70 different primary transgenic lines from each construct in short days. About 96% of rSPL9-APKVA (T-A), 65% of rSPL9-TPKVT (T-T), and 33% of rSPL9-DPKVD (T-D) independent primary transformants produced abaxial trichomes on leaf 1, respectively (Fig. 5d), suggesting that constitutive phosphorylation of SPL9 significantly impaired its function to promote vegetative phase change, while non-phosphorylated form of SPL9 enhanced the frequency of transgenic plants with strong phenotype. Next, we generated homozygous lines with a single T-DNA insertion in progenies from these transgenic lines, and chose some representative homozygous lines with similar or comparable levels of the SPL9 transcript to determine the corresponding level of the SPL9 protein and to characterize their phenotype. Western blotting result indicated that rSPL9-APKVA (T-A) plants had the highest level of SPL, rSPL9-TPKVT (T-T) had the intermediate level of SPL9, while rSPL9-DPKVD (T-D) had the lowest level of SPL9 (Fig.5g). Detailed phenotypic characterization of different representative lines indicated that rSPL9-DPKVD (T-D) transgenic plants produced much later abaxial trichomes on leaves 3-5 with much rounder first leaves (Fig. 5e, h). Within rSPL9-APKVA(T-A) and rSPL9-TPKVT(T-T) transgenic plants that produced abaxial trichomes on leaf 1, rSPL9-APKVA(T-A) produced the first leaves with slightly bigger L/W ratio than rSPL9-TPKVT(T-T) did (Fig. 5e, h). Based on our results, we think the protein levels of SPL9 in different transgenic lines of rSPL9-APKVA(T-A), rSPL9-TPKVT(T-T), and rSPL9-DPKVD(T-D) are correlated with the vegetative phase change phenotype. The correlation between rSPL9-APKVA(T-A) and rSPL9-TPKVT(T-T) seems a bit smaller, this is probably attributable to the fact the both genotypes already generated an extreme phenotype with abaxial trichomes on leaf 1. These results imply that BIN2-mediated phosphorylation of the TPKVT motif in the conserved SBP domain is critical for SPL9 function to promote vegetative phase change by destabilizing the SPL9 protein and sequestering its function. Our new experiment with the phosphorylated and non-phosphorylated forms of TOE1 also suggest that there is a correlation between the level of TOE1 and the corresponding phenotype in transgenic plants (Fig.7a-e).

5. The authors mentioned that the bin2-3 bil1 bil2 triple mutant also displays a delayed vegetative phase change phenotype, which is opposite to the expectation and the working model. To resolve the contradiction, the levels of phosphorylation and protein levels of SPL9 need to be determined.

Response: We appreciate this constructive comment. According to our model that BRs promote vegetative phase change, we expect that *bin2-3bil1bil2* would exhibit a precocious vegetative phase change phenotype. However, *bin2-3bil1bil2* also exhibits a delayed vegetative phase

change phenotype. Puzzled by this inconsistency and inspired by the comments raised by both reviewers, we first determined the level of the SPL9 protein in Ws (Wild type, *bin2-3bil1bil2* is in the Ws background) and *bin2-3bil1bil2* using Arabidopsis protoplast transformed with *pSPL9::3xFLAG-rSPL9*, and the results showed that SPL9 was significantly elevated in *bin2-3bil1bil2* (Supplementary Fig.2b). This is consistent with the idea that abolishment of the GSK3 kinase would lead to low levels of phosphorylated SPL9 to increase its stability. To identify BIN2 phosphorylation sites in SPL9, we performed liquid chromatography-tandem mass spectrometry (LC-MS/MS) analysis. LC-MS/MS analysis of the phosphorylated 3xFLAG-SPL9 in Ws and *bin2-3bil1bil2* showed that the threonine at the 102nd position in SPL9 was phosphorylated in Ws, but not in *bin2-3bil1bil2*, and that position happens to locate at the GSK3 kinase classical phosphorylation motif of (T/S)-X-X-X-(T/S) (Fig.5a, Supplementary Fig. 2a). Since our new experiments showed that BIN2 also physically interacts with TOE1 to phosphorylate it (Fig.6, 7); therefore, we studied the phosphorylation status of TOE1, and determined the level of TOE1 in *bin2-3bil1bil2*. LC-MS analysis demonstrated that TOE1 is only phosphorylated in Ws, but not in *bin2-3bil1bil2* (Supplementary Fig. 4a). As expected, the level of the miR172-insensitive form of TOE1 was elevated by about 2-fold in *bin2-3bil1bil2*, whereas the miR172-sensitive form of TOE1 is slightly reduced (Supplementary Fig. 4b, c). It is possible that the elevated level of SPL9 in *bin2-3bil1bil2* activates miR172 expression to further downregulate TOE1. We can anticipate that a reduced level of TOE1 would yield a precocious vegetative phase change phenotype; however, vegetative phase change was delayed in *bin2-3bil1bil2*. Therefore, the delayed vegetative phase change phenotype of *bin2-3bil1bil2* could not be explained by its effect on the level of TOE1. Further analysis indicated that the level of *GL1*, one of the TOE1 targets and a positive regulator of leaf epidermal trichome development, was significantly downregulated in *bin2-3bil1bil2*, while other genes in the miR156-SPL-miR172 pathway remains comparable between Ws and *bin2-3bil1bil2* (Supplemental Fig. 3b). This result implies that although the GSK3 kinase family functions to regulate vegetative phase change by phosphorylating SPL9 and TOE1, two key players in vegetative phase change, this family also functions through unidentified pathways to affect genes downstream of TOE1.

Reviewer #2 (Remarks to the Author):

In this manuscript, the authors examined the function of plant steroid hormone, Brassinosteroid (BR), in the regulation of juvenile-to-adult transition (JAT) in Arabidopsis. They provided genetic and molecular evidence that BR functions through BIN2 kinase to regulate SPL9/miR172B/TOE1/TOE2 to regulate juvenile to adult transition in a pathway:

**BR—I BIN2—I SPL9 ->miR172B—I TOE1 TOE2—I Juvenile-to-Adult Transition (JAT).
(see attached)**

The study is of potential interests to the field, but some of the studies regarding molecular mechanisms are not well established. The current model cannot explain several evident genetic results. I have some questions and suggestions for the authors to address and improve the manuscript.

1. Why BR treatment has no effect on JAT in WT (Fig 1g)?

Response: Thanks for this good question. Our study showed that 10 nM BL treatment had almost no effect on WT phenotypes, whereas BL treatment of *dwf5* plants slightly but significantly accelerated the abaxial trichome production (*dwf5*-MOCK 10.1±0.6 versus *dwf5*-10nM BL 9.0±0.5). To exclude the possibility that the addition of BL did not work for WT plants, we characterized the root phenotype in response to different concentrations of BL. Our results showed that BL treatment reduced the difference in root length between WT and *dwf5* (Fig.3), suggesting that BL treatment worked as expected. We also studied the effect of different concentrations of BL treatment on the vegetative phase change phenotype of WT and *dwf5*, we only saw that addition of 10 nM BL can partially rescue the vegetative phase change phenotype in *dwf5*, but not in WT. We initially thought the reason that BL treatment has no effect on vegetative phase change phenotype is due to the possibility of low absorption of BL by the above-ground plant parts or WT already accumulates enough BRs required for normal development, a further increase in BRs will not alter its normal development. This assumption is also consistent with other studies that BL treatment of mutants in the BR biosynthetic or signaling pathway could only restore root, hypocotyl length, and pedicel to the wild-type level, but not the silique phenotype (Li, et al., BIN2, a new brassinosteroid-insensitive locus in *Arabidopsis*. *Plant Physiol*, 2001, 127, 14-22; He, et al. BZR1 is a transcriptional repressor with dual roles in brassinosteroid homeostasis and growth responses. *Science*, 2005, 307, 1634-1638; Choe, et al. The *Arabidopsis dwf7/ste1* mutant is defective in the delta7 sterol C-5 desaturation step leading to brassinosteroid biosynthesis. *Plant Cell*, 1999, 11,207-221). Our further work suggests that BIN2 functions in a dual manner to interact with SPL9 and its downstream protein TOE1 simultaneously with opposite outcomes for TOE1 as shown in the main text. We think this mode of BIN2 action probably also contributes to the vegetative phase change phenotype we saw for WT and *dwf5* subjected to BL treatment.

Fig.3. Effect of BL treatment on the phenotype of WT and *dwarf5*. The root phenotype (a) and vegetative phase change phenotype (b) in response to BL treatment. Asterisks indicate significant difference from MOCK-treated *dwarf5* ($P < 0.01$).

2. In Fig 2b, MIR172B is decreased, why TOE1 and TOE2 are not increased as MIR172B targets?

Response: We appreciate this good question raised by both reviewer which we did not described clearly in the previous version of the manuscript. As shown in Xuemei Chen's work published in 2004, miR172 regulates *AP2*-like target gene expression primarily through translational inhibition (Chen Xuemei, A microRNA as a translational repressor of *APETALA2* in Arabidopsis flower development. *Science*, 2004, 303: 2022-2025.). Therefore, it is not surprising that *TOE1* and *TOE2* transcripts are not affected when miR172 level is reduced. Actually, our experiment showed that *TOE1* protein was significantly upregulated in *dwarf5* (Fig. 8b in the main text, also shown below in Fig.4).

Fig.4. The level of miR172-sensitive form of TOE in *dwf5* (also shown in the main text in Fig.8a)

3. The author showed SPL9 is less stable in *dwf5* mutant in Fig 2c/e, which should be confirmed by cycloheximide (CHX) treatment experiments.

Response: To confirm that SPL9 is less stable in *dwf5*, we treated *pSPL9::3×FLAG-rSPL9* plants with MG132 and/or cycloheximide (CHX). As expected, the SPL9 protein level was lower in *dwf5* than that in WT in the DMSO-treated sample, CHX treatment alone led to a further reduction in SPL9 in *dwf5* than in WT. However, MG132 treatment along or in combination with CHX inhibited SPL9 degradation (Fig. 2e in the main text, also shown below in Fig.5). These results suggest that SPL9 is subjected to proteasome-dependent degradation, and it is much less stable in *dwf5* than in WT.

Fig.5. The SPL9 protein is less stable in *dwf5* (also shown in the main text in Fig.2e)

4. The accumulation of SPL9 by BL treatment shown in Fig 1f is minimal (1.18X) and should be done with a time course experiment, for example with 0.5, 1, 2, 4 and 8 hr BL treatment.

Response: We appreciate this constructive comment. We harvested *dwf5* plants treated with BL at different time points. Western blotting showed that exogenous BR treatment led to significant SPL9 accumulation in *dwf5* as shown in Fig.2f (also shown below in Fig.6).

Fig.6. BL treatment increased the level of the SPL9 protein at different time points in *dwf5* (also shown in the main text in Fig.2f)

5. The epistasis analysis in Fig 3 is very important and well done, but the description is not clear. In principle, overexpression of SPL9, MIR172B and loss-of-function *toe1 toe2* mutant all suppressed *dwf5* mutant phenotype in JAT, supporting the linear model I summarized above. Please clearly describe these nice genetic results.

Response: Thanks for this suggestion. Our genetic analysis indicates that *toe1 toe2* could fully restore the phenotype of *dwf5* with respect to abaxial trichome production. miR172 overexpression in *dwf5* could also significantly rescue the late vegetative phase change phenotype of *dwf5*. These results suggest that miR172 functions downstream of *DWF5* with respect to leaf epidermal trait development during vegetative phase change, and the reduction of miR172 in *dwf5* is partially responsible for the late abaxial trichome phenotype of *dwf5*. In contrary to distinct abaxial trichome phenotypes as manifested by *dwf5*, *Ubi10::172B dwf5*, and *toe1 toe2 dwf5*, leaf shape of *Ubi10::172B dwf5* and *toe1 toe2 dwf5* resembled that of *dwf5* to a greater extent. This result suggests that *dwf5* is epistatic to miR172 and *TOE1/TOE2* with respect to leaf shape development during vegetative phase change, and BRs contribute to leaf shape development by affecting genes other than miR172 and *TOE1/TOE2*. This is also in accordance with our previous result that miR172 and *TOE1/TOE2* only contribute to leaf epidermal trait development in vegetative phase change. Please refer to the revised paragraph in this section.

6. The phosphorylation of SPL9 by BIN2 shown in Fig 5 should be more vigorously established. The kinase assay should be done with isotope for better sensitivities. The mutant SPL9 protein with two putative BIN2 site changed to Ala should be tested in kinase assay with BIN2.

Response: We fully appreciate this advice. To vigorously establish that SPL9 is phosphorylated by BIN2, we first performed a liquid chromatography-tandem mass spectrometry (LC-MS/MS) analysis and identified that the threonine at the 102nd position in the SPL9 protein sequence is phosphorylated, then we mutated the threonine to an alanine, and purified the bacterium-expressed GST-BIN2, 6×His-SPL9 (T) and 6×His-SPL9 (A). Last, we performed *in vitro* kinase assays using γ -³²P-ATP. The result indicates that BIN2 can phosphorylate SPL9 *in vitro* (Fig.5c in the main text, also shown below in Fig.7). Because our new experiments showed that BIN2 can also interact with TOE1. Therefore, we also performed *in vitro* kinase assays for TOE1 using γ -³²P-ATP. Again, the result indicated that BIN2 can phosphorylate TOE1 (Fig.7a in the main text, also shown below in Fig.7).

Fig.7. SPL9 and TOE1 are phosphorylated by BIN2 (also shown in the main text in Fig.5c, Fig.7a)

7. The effect of SPL (T-A) and SPL9 (T-D) mutations should be examined for A and D forms in terms of protein stability (Fig 5g). The author should identify transgenic lines with similar transcript levels to examine the protein levels and stability (i.e. CHX treatment). It's likely that T-A mutation stabilizes and T-D mutation destabilize the SPL9 proteins.

Response: We appreciate this constructive comment raised by both reviewers. Based on this suggestion, we characterized the vegetative phase change phenotype of more than 70 different primary transgenic lines transformed with each construct in short days. About 96% of rSPL9-APKVA (T-A), 65% of rSPL9-TPKVT (T-T), and 33% of rSPL9-DPKVD (T-D) plants produced abaxial trichomes on leaf 1, respectively (Fig. 5d), suggesting that constitutive phosphorylation of SPL9 significantly impaired its function to promote vegetative phase change. To understand how SPL9 phosphorylation affects vegetative phase change phenotype and the protein levels, we first generated homozygous lines with a single T-DNA insertion in progenies from these transgenic lines, and determined the transcript levels of SPL9 in different homozygous lines. We then characterized the phenotype of these lines with comparable SPL9 transcript levels (Fig.5e, f). rSPL9-DPKVD (T-D) transgenic plants produced much later abaxial trichomes on leaves 3-5 with much rounder first leaves (Fig. 5e, h). Within rSPL9-APKVA(T-A) and rSPL9-TPKVT(T-T) transgenic plants that produced abaxial trichomes on leaf 1, rSPL9-APKVA(T-A) produced the first leaves with slightly bigger L/W ratio than rSPL9-TPKVT(T-T) did (Fig. 5e, h). To see how phosphorylation affects SPL9 accumulation and the phenotype, we performed Western blotting using an anti-FLAG antibody to determine the levels of the SPL9 protein in these transgenic lines with similar/comparable SPL9 transcript (Fig.5f). Western blotting indicated that rSPL9-APKVA(T-A) produced the highest SPL9 level, rSPL9-TPKVT(T-T) had the intermediate level of SPL9, while rSPL9-DPKVD (T-D) had the lowest level (Fig.5g). The protein level of SPL9 is also in line with the phenotype we observed (Fig.5d,e,h). These results imply that the status of BIN2-mediated phosphorylation of the TPKVT motif in the conserved SBP domain is critical for SPL9 stability and function to promote vegetative phase change. Because BIN2 also phosphorylates TOE1(Fig.7a), we performed similar experiments for TOE1. The result also suggests that transgenic plants with rTOE1-APKVA (T-A) produced the strongest phenotype with the highest level of TOE1, rTOE1-TPKVT (T-T) had an intermediate phenotype and an intermediate level of TOE1, while rTOE1-DPKVD (T-D) had the weakest phenotype and the lowest level of TOE1 (Fig.7b-e)

8. The *bin2* loss-of-function and *bzr-1D* gain-of-function mutants have opposite phenotypes to what can be predicated from the current model. The authors did not provide a viable explanation. One possibility is that TOE1 and TOE2 genes are induced by BR (RNA-seq data i387n Wang et al., *Molecular Plant*. 7: 1303-1315). So it's possible that BR affects JAT through two branched pathways with opposite outcomes likely under different conditions (one through BIN2-SPL9/MIR172B/TOE1/2 and another through BIN2-BES1/BZR1/TOE1/2, see attached ppt). This can explain the two mutant phenotypes here as well as answer questions I raised in points 1 and 2. I hope the authors can test this possibility by examining TOE1 and TOE2 gene expression in various mutants.

Response: Thanks for this very important suggestion. Western blotting analyses indicated that SPL9 was elevated, but the miR172-sensitive TOE1 was reduced in *bin2-3bil1bil2* (Supplementary Fig. 2b, 4c). This result contradicts the delayed vegetative phase change phenotype of *bin2-3bil1bil2*. Our further experiments showed that BIN2 also interacts with and phosphorylates TOE1 (Fig.6,7), and the BIN2-TOE1 regulatory mode is important to keep TOE1 from overaccumulation (Supplementary Fig. 4b) because the miR172-insensitive TOE1 was elevated in *bin2-3bil1bil2*. In *dwf5*, the miR172-sensitive TOE1 was elevated, but the miR172-insensitive TOE1 was reduced (Fig.8a,b). Therefore, BIN2 functions in a dual manner to regulate vegetative phase change with opposite outcomes: one is to interact with SPL9 physically to destabilize it, thus reducing the level of miR172 to increase the TOE1 protein level, the other is to interact with TOE1 downstream of SPL9 to destabilize it. Based on these results, we proposed a new model for the interaction of BRs with vegetative phase change in *Arabidopsis* (Fig.8c). These new data confirmed the idea raised by the reviewer that BRs affect vegetative phase change through two branched pathways with opposite outcomes for TOE1. The complicated role of the GSK3 kinase family and *BZR1* in vegetative phase change still awaits further investigation. Please refer to our main text for this part.

Reviewer #1 (Remarks to the Author):

The authors have seriously evaluated both reviewer's comments and performed series of new experiments to carefully address the concerns. The quality of the MS is much improved and conclusions much solidified. I am satisfied with the revision.

Reviewer #2 (Remarks to the Author):

The authors carefully considered my comments and adequately addressed all of them. I am very glad that the authors took time to perform many new experiments, some of which are rather challenging to do, to support the main conclusions. As someone worked in BR field for more than 20 years, I wanted to point out that in our own research, we also found that bin2 triple loss-of-function mutant (bin2-3 bil1 bil2) in some cases has opposite phenotypes as expected. One possibility, as the authors indicated, is that BIN2 and its homologs may have other substrates that contributed to the complex phenotypes to fine tune a specific biological process. The results not only established the mechanism of BR regulation of juvenile to adult transition, but also provide insight into the intriguing feature of BIN2 regulation of biological processes (i.e. dual regulation of BIN2 on SPL9 and TOE1 with opposite outcomes in the pathway).